# Systematic identification of cancer cell vulnerabilities to natural killer cell-mediated immune surveillance

**Matthew F Pech, Linda E Fong, Jacqueline E Villalta, Leanne JG Chan, Samir Kharbanda, Jonathon J O'Brien, Fiona E McAllister, Ari J Firestone, Calvin H Jan, Jeffrey Settleman\***

Calico Life Sciences LLC, South San Francisco, United States

**Abstract** Only a subset of cancer patients respond to T-cell checkpoint inhibitors, highlighting the need for alternative immunotherapeutics. We performed CRISPR-Cas9 screens in a leukemia cell line to identify perturbations that enhance natural killer effector functions. Our screens defined critical components of the tumor-immune synapse and highlighted the importance of cancer cell interferon-$\gamma$ signaling in modulating NK activity. Surprisingly, disrupting the ubiquitin ligase substrate adaptor DCAF15 strongly sensitized cancer cells to NK-mediated clearance. DCAF15 disruption induced an inflamed state in leukemic cells, including increased expression of lymphocyte costimulatory molecules. Proteomic and biochemical analysis revealed that cohesin complex members were endogenous client substrates of DCAF15. Genetic disruption of DCAF15 was phenocopied by treatment with indisulam, an anticancer drug that functions through DCAF15 engagement. In AML patients, reduced DCAF15 expression was associated with improved survival. These findings suggest that DCAF15 inhibition may have useful immunomodulatory properties in the treatment of myeloid neoplasms.

DOI: https://doi.org/10.7554/eLife.47362.001

**\*For correspondence:**
jsettleman@gmail.com

## Introduction

Major advances in tumor control have recently been achieved by targeting immune inhibitory signaling pathways. Treatment with 'checkpoint inhibitors,' antibodies targeting PD1, PD-L1, or CTLA4, lead to durable responses across a wide range of indications, but only in a subset of patients. Treatment response is positively correlated with tumor mutational burden and infiltration of CD8+ effector T cells, which recognize tumor cells via peptides bound to major histocompatibility complex class I (MHC-I) molecules, suggesting that checkpoint inhibitors work best at clearing highly immunogenic cancers with repressed T cell responses (*Snyder et al., 2014*; *Tumeh et al., 2014*; *Mariathasan et al., 2018*). Substantial efforts are being made to extend the benefits of immunotherapy to additional patients, including combining checkpoint inhibitors with other therapies, drugging additional lymphocyte-suppressive pathways, and boosting the activity of other arms of the immune system (*Galon and Bruni, 2019*).

Resistance to therapy has long been a major problem in cancer treatment. Drugs targeting tumor growth pathways can profoundly reduce tumor burden, but resistance invariably arises, driven by the substantial genetic and phenotypic heterogeneity present within human tumors (*Easwaran et al., 2014*). Recent clinical and experimental data have similarly highlighted the ability of cancer cells to escape checkpoint inhibitor-induced immune control. B2M and JAK1/2 mutations have been identified in melanoma patients with acquired resistance to checkpoint inhibitors (*Zaretsky et al., 2016*; *Sade-Feldman et al., 2017*). These mutations impair recognition of the tumor by the adaptive immune system, either by directly disrupting antigen presentation or by

**eLife digest** The human immune system can recognize and kill cancer cells growing in the body. Certain immune cells recognize mutated proteins on the surface of cancer cells known as antigens, and this ability can be enhanced by drugs. These so-called immunotherapies can be effective to treat several cancer types, but only some patients benefit from them. This is because cancer cells often stop presenting antigens on their surface, thus hiding from the immune response.

Natural killer cells are a type of immune cell that does not rely on antigen presentation to recognize cancer cells. Scientists are now trying to develop drugs to increase the effectiveness with which natural killer cells attack cancer.

Pech et al. used cells from a human leukemia, a type of blood cancer, to look for proteins that made these cells more vulnerable to natural killer cells. The main experiment, in which every single protein in the cancer cells was deleted one by one, revealed that a protein called DCAF15 changes how cancer and natural killer cells interact. Leukemia cells lacking DCAF15 could be attacked by natural killer cells much more easily because the cancer cells exhibited inflammation-like symptoms that stimulated the immune response. DCAF15 is part of a family of 'adaptors' that that provide specificity to the cellular machinery that controls proliferation, the recycling of proteins and DNA repair.

Inhibiting DCAF15 with a drug also made natural killer cells more efficient at eliminating leukemia cells. Patients with leukemia whose cancer cells make little DCAF15 protein have a better chance of survival, suggesting that this process may already be happening in some patients.

Together these data indicate that targeting DCAF15 in leukemia patients may help natural killer cells attack cancer cells. Future research is needed to see if a similar process takes place in other cancer types.

DOI: https://doi.org/10.7554/eLife.47362.002

rendering the cells insensitive to IFNγ, an important inducer of MHC-I expression. Functional genetic screens using T cell-cancer cell cocultures have highlighted similar mechanisms of resistance in vitro (*Kearney et al., 2018*; *Pan et al., 2018*; *Manguso et al., 2017*; *Patel et al., 2017*). Even treatment-naïve tumors can be highly immuno-edited, presenting with IFNγ pathway mutations, reduced MHC-I expression and loss of the peptide sequences that can serve as antigens (*Dunn et al., 2004*; *Rodig et al., 2018*; *Gao et al., 2016*; *McGranahan et al., 2017*). Together, these findings highlight a critical need for therapies that can either increase MHC expression or work in a MHC-independent fashion.

Anti-tumor immunity is not solely mediated by the adaptive immune compartment. Innate immune cells, most notably natural killer (NK) cells, can have both direct tumoricidal activity and also help to fully elaborate long-lasting anti-tumor responses (*Marcus et al., 2014*; *Moynihan et al., 2016*; *Lopez-Soto et al., 2017*; *Chiossone et al., 2018*). NK cells are cytotoxic lymphocytes capable of mounting rapid responses to damaged, infected, or stressed cells, including cancer cells. T and NK cells share effector functions, releasing cytokines and exocytosing lytic granules upon activation to kill target cells. However, NK activation status is controlled by the integrated signals from germ-line-encoded NK-activating and -inhibiting receptors (aNKRs/iNKRs). Generally, iNKR ligands are expressed by normal and healthy cells, whereas aNKR ligands are upregulated after DNA damage or viral insult (*Chiossone et al., 2018*; *Ljunggren and Malmberg, 2007*). MHC-I molecules provide a potent inhibitory signal sensed by NK cells, enabling the innate immune system to respond productively to MHC-deficient cells. As a result, there is considerable interest in amplifying NK responses to cancers, as well as developing NK-based cell therapies (*Lopez-Soto et al., 2017*; *Chiossone et al., 2018*; *Ljunggren and Malmberg, 2007*).

Here, we performed genetic screens in an MHC-deficient leukemic cell line to systematically identify modulators of NK-mediated anti-cancer immunity. These screens, unexpectedly, revealed the potential therapeutic utility of targeting the cullin4-RING E3 ubiquitin ligase (CRL4) substrate adaptor DCAF15 in myeloid malignancies. Disruption of DCAF15 strongly sensitized cancer cells to NK-mediated killing, resulting from increased cancer cell expression of lymphocyte costimulatory molecules. Proteomic experiments revealed that DCAF15 interacted with and promoted the ubiqitination

of the cohesin complex members. Treatment with indisulam, an anticancer drug that modulates DCAF15 function, reduced interaction with cohesin members and mimicked DCAF15 loss-of-function immunophenotypes.

## Results

### A genome-scale CRISPR screen identifies modulators of NK effector functions

We performed genome-scale CRISPR screens in K562 cells to identify perturbations that modulate NK-92-mediated killing (*Figure 1A*). K562 human chronic myelogenous leukemia cells are a NK-sensitive cancer cell line that weakly expresses MHC-I. For screening purposes, a clonal isolate of K562 cells expressing high levels of spCas9 was generated and validated (*Figure 1—figure supplement 1A–C*). NK-92 cells are a human lymphoma-derived cell line phenotypically similar to activated NK cells (*Klingemann et al., 1994*). These cells exhibit interleukin-2 (IL-2)-dependent growth, express a large number of aNKRs and few iNKRs (*Maki et al., 2001*), and display potent cytolytic activity against K562 cells.

In pilot experiments, labeled K562 cells were co-cultured with NK-92 cells to determine an effector-to-target (E:T) ratio that applied sufficient selective pressure for screening (*Figure 1—figure supplement 1D–F*). IL-2 was removed during the co-culture to promote the eventual death of NK-92 cells, allowing the collection of genomic DNA preparations undiluted by effector cell DNA. A multi-day timeframe between the NK-92 challenge and screen readout was used to capture tumor cell fitness changes related both to the direct cytolytic activities of NK cells as well as the longer-term effects from NK-released cytokines.

For the co-culture screen, cas9-expressing K562 cells were infected with a genome-scale single guide RNA (sgRNA) library targeting all unique coding genes and miRNAs, as well as one thousand non-targeting controls (*Supplementary file 1*). Seven days post-infection, cells were either grown normally or challenged with NK-92 cells at a 1:1 or 2.5:1 E:T ratio, reducing K562 cell counts 19-fold or 43-fold, respectively, by the end of the screen (*Figure 1B*). Deep sequencing was used to compare changes in sgRNA abundance between the challenged and unchallenged state after 8 days of co-culture, and genes were ranked using the MAGeCK software (*Li et al., 2014*). (*Figure 1C*, *Supplementary file 2*) There was good agreement between the results from screens performed at the different E:T ratios (*Figure 1—figure supplement 2*).

The screen revealed two broad classes of 'hits'— sgRNAs targeting components of the tumor-immune synapse or components of the IFNγ signaling pathway (*Figure 1C–D*). Disruption of ICAM1 was the top-ranked NK-92 evasion mechanism, scoring many orders of magnitude stronger than any other gene—an observation consistent with the critical role of ICAM1-LFA1 interactions in establishing initial target-lymphocyte adhesion and polarizing cytotoxic granules towards the synapse (*Marcus et al., 2014*). Single guide RNAs targeting multiple other tumor-immune synapse components were also enriched after NK-92 challenge, including NCR3LG1 (#26-ranked gene by MaGeCK score), the activating ligand for NKP30 on NK cells (*Brandt et al., 2009*); CD58 (#37), an adhesion molecule that binds CD2 (*Selvaraj et al., 1987*; *Rölle et al., 2016*); and CD84 (#80), a SLAM-related receptor that binds homotypically to promote activation and cytokine secretion in lymphocytes (*Martin et al., 2001*; *Veillette, 2006*; *Wang et al., 2010*). Other than NECTIN2, sgRNAs targeting NK-inhibitory surface proteins did not score prominently as NK-sensitization mechanisms, consistent with the weak MHC-I expression on K562 cells and the limited repertoire of NK inhibitory receptors expressed on NK-92 cells (*Maki et al., 2001*). NECTIN2 transmits both stimulatory or inhibitory signals to NK cells, depending on whether it is bound to DNAM1 (CD226) or TIGIT, respectively (*Stanietsky et al., 2009*; *Bottino, 2003*).

After ICAM1, the top 10 highest scoring NK-92 evasion mechanisms were dominated by sgRNAs targeting the proximal components of the IFNγ signaling pathway, including STAT1, JAK1, IFNGR2, JAK2 and INFGR1 (*Figure 1D*). Consistent with the importance of cancer cell IFNγ signaling, sgRNAs targeting negative regulators of the interferon response were strongly depleted after NK-92 challenge. Disruption of the protein tyrosine phosphatases PTPN2 and PTPN1 were the #2 and #5 ranked NK-92 -sensitizing mechanisms, respectively. Presumably, these proteins suppress IFNγ-induced immunomodulation by dephosphorylating STAT and JAK proteins, as has been reported in

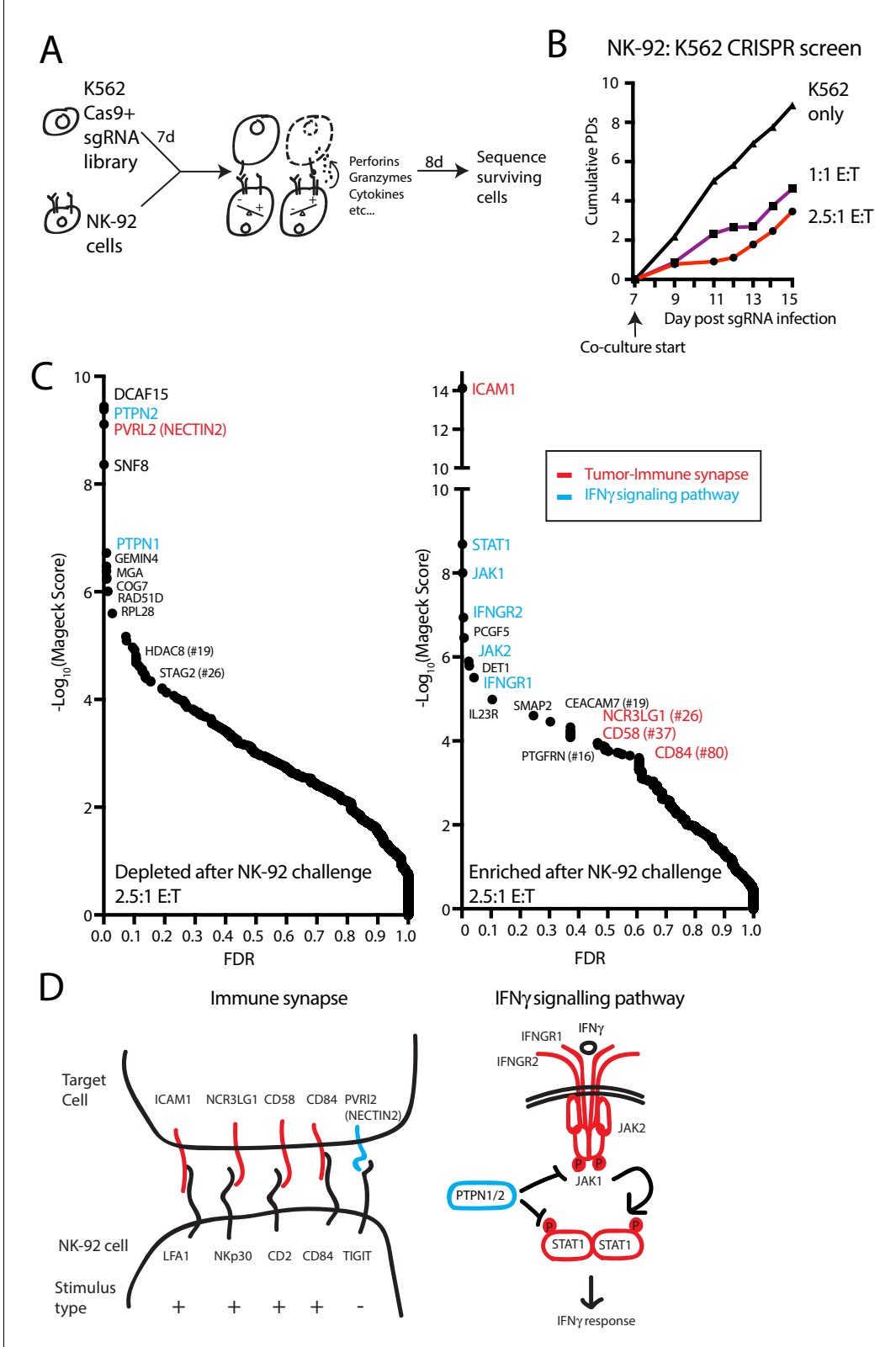

**Figure 1.** A genome-scale CRISPR screen identifies modulators of NK effector functions. (**A**) Overview of the genome-scale NK CRISPR screening system. (**B**) K562 population doublings (PDs) during the CRISPR screen, as measured by total number of live cells. Note that early timepoints from the co-culture reflect the presence of both K562 and NK-92 cells. (**C**) Analysis of the NK CRISPR screen results. Changes in sgRNA abundance were compared between the 2.5:1 E:T co-culture condition and day 15 dropout cells using the MAGeCK algorithm. The top 10 enriched or depleted genes

*Figure 1 continued on next page*

*Figure 1 continued*

are shown, as rank-ordered by MAGeCK score; other manually selected genes are highlighted with their rank indicated. FDR, false discovery rate. (D) Overview of high-scoring components of the tumor-immune synapse and IFNγ signaling pathway recovered by the screen. sgRNAs against genes enriched after exposure to NK-92 cells are marked in red, while depleted sgRNAs are marked in blue.

DOI: https://doi.org/10.7554/eLife.47362.003

The following figure supplements are available for figure 1:

**Figure supplement 1.** Optimization of NK CRISPR screens.

DOI: https://doi.org/10.7554/eLife.47362.004

**Figure supplement 2.** Additional NK CRISPR screen data.

DOI: https://doi.org/10.7554/eLife.47362.005

CRISPR screens using T-cell coculture systems or syngeneic tumor models (*Pan et al., 2018*; *Manguso et al., 2017*). Taken together, these findings indicate that our in vitro functional genomics screens effectively revealed known components of physiologically-relevant immune synapse and cytokine pathways.

## Prospective identification of novel genes affecting sensitization to NK cells

Mechanisms of NK-92 sensitization identified in the screen were diverse, revealing many strongly-scoring genes not previously linked to either interferon signaling or NK cell biology (*Figure 1*). Most surprisingly, the top-ranked mechanism for promoting NK-92 mediated clearance was disruption of DCAF15, an uncharacterized substrate adaptor for CRL4 ubiquitin E3 ligases. DCAF15 is a member of the large family of DDB1 and Cul4-associated factors (DCAFs) (*Jin et al., 2006*). CRL4 complexes enable cells to mark proteins for proteosomal degradation, helping regulate intracellular protein homeostasis. As substrate adaptors for CRL4, DCAF proteins provide specificity to the complex, determining which client proteins are ubiquitinated (*Jackson and Xiong, 2009*). As with most substrate adaptors, the normal client repertoire of DCAF15 is undefined, and relatively little is known about the biological function of DCAF15.

We also noted that disruption of two cohesin-related genes, STAG2 and HDAC8, scored as NK-92 sensitization factors (ranked #26 and #19, respectively). Cohesin is a ring-shaped complex involved in chromatin replication, organization and repair, with STAG2 acting as a core complex member and HDAC8 controlling chromatin accessibility (*Uhlmann, 2016*). Cohesin dysregulation has cell context-specific consequences, including DNA damage and aneuploidy; in leukemic cells, cohesin mutations are thought to enforce stem cell programs by altering chromatin organization (*Mazumdar and Majeti, 2017*).

## A phenotypic screen based on MHC-I upregulation to identify modulators of the IFNγ response

The prominent role for IFNγ signaling in the immune response to cancer cells, both clinically and in our screens, prompted us to define more specifically which NK-92-sensitizing genes are involved in modulating the IFNγ response. MHC-I levels are highly upregulated in K562 cells after IFNγ exposure, increasing 5.9+ /- 0.98 fold after 24 hr of exposure to IFNγ. This induction was dependent on STAT1 and was nearly doubled by disrupting PTPN2 (*Figure 2A–B* and *Figure 2—figure supplement 1*). We therefore used IFNγ-induced cell surface MHC-I expression as a proxy for the strength of the interferon response. K562 cells transduced with a genome-scale CRISPR library were treated with IFNγ for 24 hr and MHC-I expression was measured by flow cytometry. The brightest 20% and dimmest 20% of cells were sorted, and deep sequencing was used to compare sgRNA abundance between the populations (*Figure 2C*, *Supplementary file 3*).

As expected, cells with impaired MHC-I upregulation were highly enriched for sgRNAs targeting the IFNγ-JAK-STAT pathway (IFNGR1/2, JAK1/2, STAT1, IRF1/2), as well as the antigen processing/presentation machinery (B2M, TAP1/2, TAPBP, PDIA3, HLA-C/B) (*Figure 2D*). Conversely, disruption of PTPN2 or STAG2 induced an exuberant MHC-I response. Surprisingly, sgRNAs targeting epigenetic factors were highly enriched within the brightest MHC-I expressing cells—most prominently,

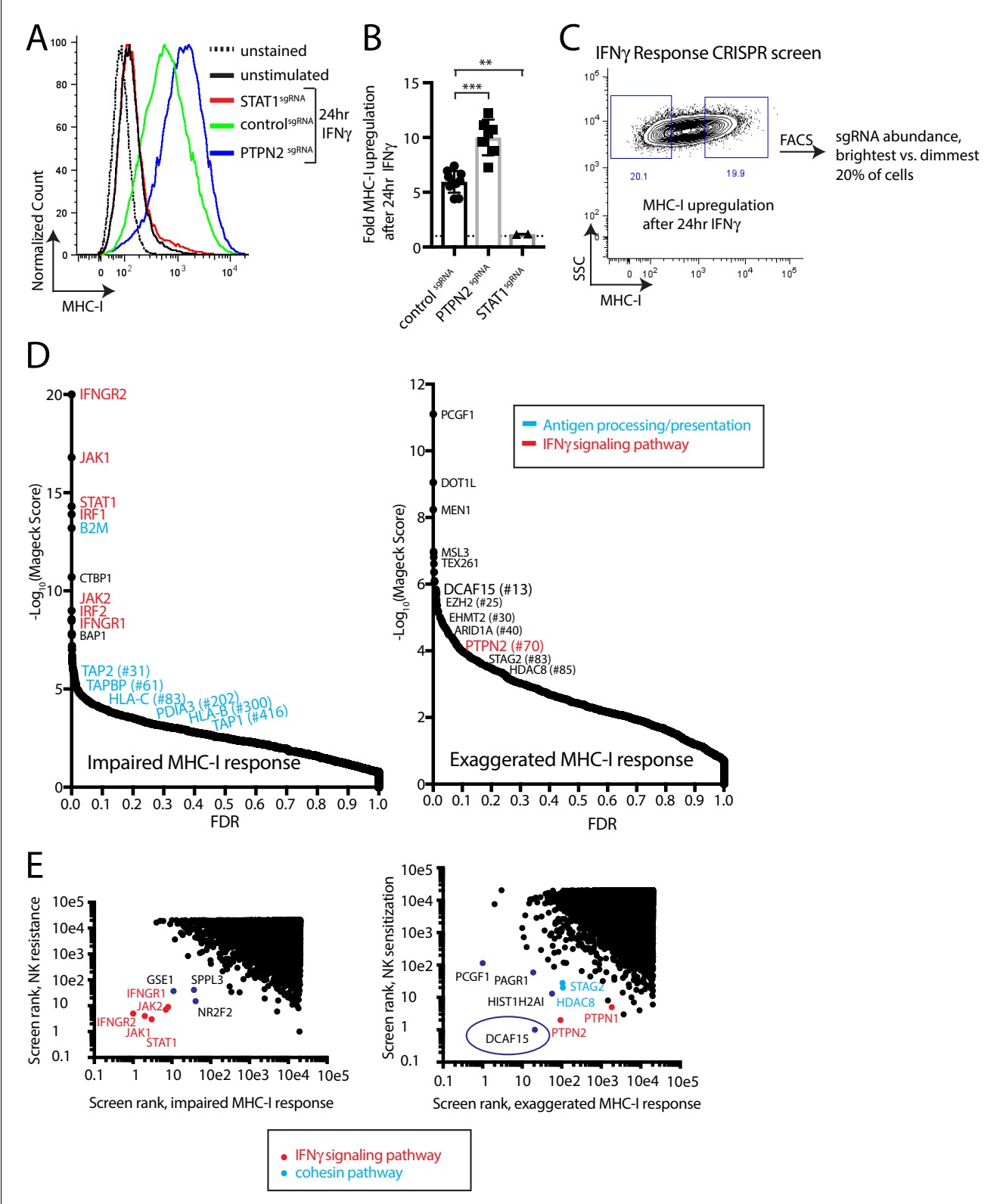

**Figure 2.** A phenotypic screen based on MHC-I upregulation to identify modulators of the IFNγ response. (**A**) Flow cytometry measurement of MHC-I expression in K562 cells transduced with the indicated sgRNAs after 24 hr of 10 ng/ml IFNγ treatment. (**B**) Fold upregulation of MHC-I expression after IFNγ treatment in K562 cells transduced with the indicated sgRNAs. Mean and standard deviation are shown. ***p value=0.0002, **p value=0.03, Mann-Whitney test. (**C**) Design of CRISPR screen for IFNγ-induced upregulation of MHC-I expression. SSC, side-scatter. (**D**) Analysis of the MHC-I

*Figure 2 continued on next page*

*Figure 2 continued*

upregulation CRISPR screen results. The MAGeCK algorithm was used to compare sgRNA abundance between cells in the bottom two and top two deciles of MHC-I expression. The false discovery rate (FDR) is plotted against the -log10 transformation of the MAGeCK score. The top 5 to 10 enriched or depleted genes are shown, as rank-ordered by MAGeCK score; other manually selected genes are highlighted with their rank indicated in parentheses. (E-F) Comparison of the NK and MHC screening results. Results of each screen were rank-ordered based on their MAGeCK score. Select genes are highlighted.

DOI: https://doi.org/10.7554/eLife.47362.006

The following figure supplement is available for figure 2:

**Figure supplement 1.** Confirmation of gene disruption in K562 cells.

DOI: https://doi.org/10.7554/eLife.47362.007

members of the BCOR complex PCGF1 and KDM2B, members of the PRC2 complex EZH2 and SUZ12, as well as factors affecting histone methylation/acetylation status.

Rank-rank comparisons between the NK and MHC screens were informative in prospectively defining a core group of IFNγ response genes in K562 cells (*Figure 2E*). Comparing sgRNAs enriched after NK-92 challenge with those causing impaired MHC-I upregulation clearly delineated the known proximal components of the IFNγ signaling pathway (IFNGR1/2, JAK1/2, STAT1), and highlighted several poorly characterized genes such as GSE1, SPPL3 and NR2F2 (*Figure 2E*).

Surprisingly, comparing sgRNAs depleted after NK-92 challenge with those causing an exaggerated MHC-I response highlighted the CRL4 substrate adaptor DCAF15 most prominently, alongside the cohesin members STAG2 and HDAC8 (*Figure 2E*). As expected, negative feedback regulators of the IFNγ pathway (PTPN1 and PTPN2) were also recovered by this analysis. We focused additional studies on understanding the function of DCAF15, given its prominence in both the NK sensitization (#1 ranked hit at 2.5 E:T ratio; #12 ranked hit at 1:1 E:T ratio) and MHC upregulation (#13 ranked hit) screens.

## Disruption of the E3 ubiquitin ligase substrate adaptor DCAF15 enhances NK effector functions

To evaluate hits from the CRISPR screens, we generated individual gene knockout (KO) cell lines by lentiviral sgRNA expression, producing polyclonal cell lines with high levels of gene disruption (*Figure 2—figure supplement 1* and *Figure 3—figure supplement 1*). Fluorescently-labeled control or test KO target cell lines were subjected to competitive co-culture assays in the presence of either NK-92 or primary NK effector cells, with changes in the relative ratios of target cell types measured over time by flow cytometry (*Figure 3A*).

As expected, disrupting ICAM1 in K562 cells conferred very high levels of protection against NK-92 cells (*Figure 3B–C*; 19.7-fold enrichment). Disabling signaling downstream of IFNγ by disrupting STAT1 provided an intermediate level of resistance (2.45-fold enrichment). K562s are very sensitive to NK-mediated killing, providing a large dynamic range for detection of resistance-promoting factors, while limiting the ability of the assay to detect similarly large increases in sensitization. Nevertheless, multiple independent sgRNAs targeting DCAF15 promoted sensitization to NK-92 cells (1.6-fold depletion), with a similar degree of preferential killing observed for NECTIN2 or PTPN2 KO cells (NECTIN2: 1.95-fold depletion; PTPN2: 1.4-fold depletion).

We repeated NK-92 competitive co-culture experiments after disruption of DCAF15, PTPN2, STAT1 and ICAM1 in Daudi cells, a B2M-deficient B-cell lymphoma line (*Figure 3D*, *Figure 3—figure supplement 1D–F*). ICAM1 KO Daudi cells were highly protected against NK-92 cell killing, whereas disruption of DCAF15 or PTPN2 led to enhanced killing. In contrast to K562 cells, STAT1 disruption in Daudi cells promoted their preferential killing.

To extend these observations to primary NK cells, human peripheral NK cells were isolated from PBMCs of 6 healthy donors, activated and challenged in competitive co-cultures with various K562 KO cell genotypes (*Figure 3E* and *Figure 3—figure supplement 2*). Disruption of DCAF15 or PTPN2 promoted sensitization to primary NK cells, albeit with reduced magnitudes of effect compared to NK-92 cells (PTPN2: 1.3-fold depletion; DCAF15: 1.15-fold depletion). In 3 out of 6 donors, NK cells showed increased degranulation, as measured by cell surface CD107a expression, when challenged with DCAF15 KO cells (*Figure 3F*). ICAM1 disruption promoted resistance to NK cell attack, but only conferred partial protection (2.3-fold enrichment). The effect of STAT1 disruption

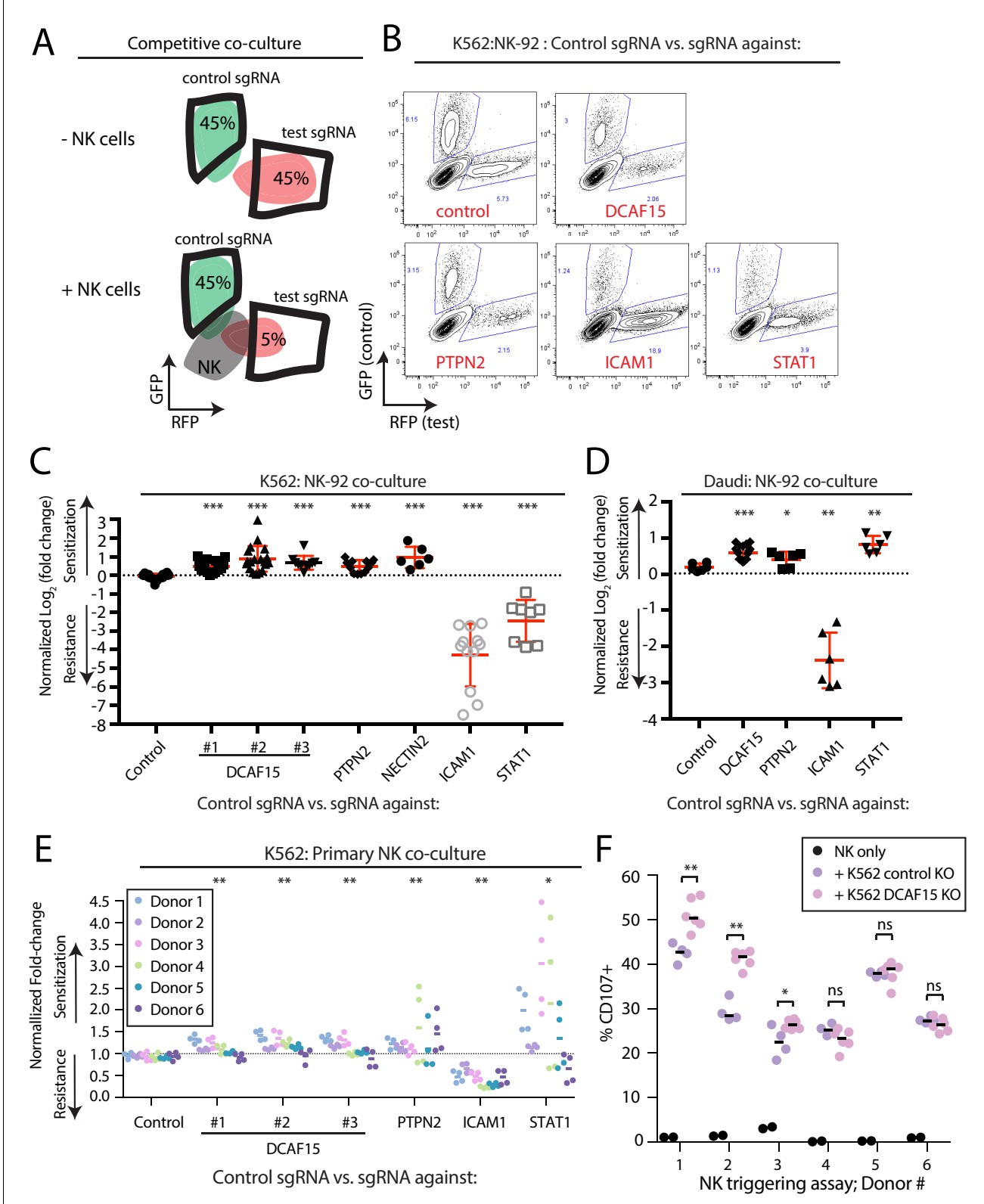

**Figure 3.** Disruption of the E3 ubiquitin ligase substrate adaptor DCAF15 enhances NK effector functions. (**A**) Experimental design of competitive co-culture experiments, with FACS data illustrating a hypothetical sgRNA that enhances NK-mediated target cell clearance. (**B**) Representative results from NK-92:K562 competitive co-culture experiments performed at a 2.5:1 E:T ratio. (**C**) Results of competitive co-culture performed at a 2.5:1 E:T ratio and measured 48–96 hr after challenge of K562 cells with NK-92 cells. Mean and standard deviation are shown. *** p-value<0.0001, Mann-Whitney test.

*Figure 3 continued on next page*

*Figure 3 continued*

(D) Results of competitive co-culture performed at a 1:1 E:T ratio and measured 48–96 hr after challenge of K562 cells with IL-2 activated isolated peripheral NK cells. ** p-value<0.01, * p-value=0.06, Wilcoxon matched-pairs signed rank test. (E) Results of competitive co-culture performed at a 2.5:1 E:T ratio and measured 48–96 hr after challenge of Daudi cells with NK-92 cells. Mean and standard deviation are shown. *** p-value<0.0001, ** p-value=0.0022, * p-value=0.09, Mann-Whitney test. F)Flow cytometry analysis of NK cell degranulation (cell surface CD107A expression) after 2 hr coculture of primary NK cells with indicated target cell types at 2.5:1 E:T ratio. Line indicates median value. ** p-value=0.095, * p-value=0.067, NS p-value>0.10, Mann-Whitney test.

DOI: https://doi.org/10.7554/eLife.47362.008

The following figure supplements are available for figure 3:

**Figure supplement 1.** Confirmation of gene disruption in K562 and Daudi cells.

DOI: https://doi.org/10.7554/eLife.47362.009

**Figure supplement 2.** Co-culture of primary NK cells with K562 cells.

DOI: https://doi.org/10.7554/eLife.47362.010

was extremely variable, with STAT1 KO K562 cells strongly preferentially killed by primary NK cells from a subset of donors. These findings implicate DCAF15 and PTPN2 as novel modulators of NK-mediated cancer cell immunity.

## DCAF15 disruption leads to an inflamed state distinct from dysregulated IFNγ signaling

DCAF15 was a strong hit in both the NK sensitization and MHC upregulation screens, suggesting that DCAF15 disruption sensitizes K562 cells to NK-mediated killing by dysregulating the IFNγ response. Consistent with the screening results, polyclonal K562 cells expressing DCAF15 sgRNAs ('DCAF15 KO cells') displayed 2.45-fold higher levels of MHC-I than control knockout cells after 24 hr of IFNγ exposure, an effect comparable in magnitude to PTPN2 disruption (*Figure 4A*). We then tested whether DCAF15 KO cells exhibited hallmarks of dysregulated JAK-STAT signaling, using PTPN2 KO cells as a positive control. We were unable to see any difference in induction of STAT1 phosphorylation after IFNγ exposure in DCAF15 KO cells, or differences in steady state levels of STAT1/2, JAK1/2 or IFNGR1 (*Figure 4B* and data not shown). DCAF15 KO cells appeared healthy and proliferated at a normal rate (*Figure 4—figure supplement 1A*). As in wild-type K562 cells, long-term IFNγ exposure was neither cytotoxic nor cytostatic to DCAF15 KO cells (Figure 4c and *Chen et al., 1986*). In contrast, PTPN2 KO cells showed higher levels of STAT1$^{PY701}$ induction after IFNγ exposure, and their proliferative rate was temporarily reduced after transient exposure to IFNγ, or more substantially slowed down by continuous treatment with the cytokine (*Figure 4B–C*).

We explored the transcriptional and immunophenotypic response of cells to IFNγ treatment. RNA-seq and flow cytometry was performed on control, DCAF15 KO or PTPN2 KO K562 cells basally and after 24 hr of IFNγ exposure. Wild-type cells dramatically upregulated transcription of anti-viral genes and components of the antigen processing and presentation pathway after IFNγ treatment (*Figure 4—figure supplement 1B* and *Supplementary file 4*). On the cell surface, K562 cells exhibited STAT1-dependent upregulation of ICAM1 expression after IFNγ treatment, with a variety of other important NK ligands unaffected by cytokine treatment (*Figure 4—figure supplement 1C*).

Clustering analysis clearly showed that PTPN2 KO cells were transcriptionally distinct from control or DCAF15 KO cells both before and after cytokine exposure (*Figure 4D*). In the basal state, PTPN2 KO cells were enriched for inflammation and interferon-associated Gene Ontology (GO) terms (*Figure 4—figure supplement 1D*). After cytokine exposure, gene set enrichment analysis revealed that PTPN2 KO cells had exaggerated transcriptional responses to interferon and were also enriched for apoptotic GO gene categories (*Figure 4—figure supplement 1E*). PTPN2 KO cells also showed greater IFNγ-induced MHC-I and ICAM1 cell surface expression (*Figure 2A* and *Figure 4—figure supplement 1C*). These results suggest that loss of appropriate IFNγ negative feedback may both promote cell death and modulate NK cell interactions.

In contrast, DCAF15 KO cells did not show substantial differences in their transcriptional response to IFNγ (*Figure 4D*). However, DCAF15 KO cells were enriched for GO terms associated with NK-mediated cytotoxicity, antigen presentation and cell adhesion, consistent with our phenotypic characterization of these cells (*Figure 4E*). Together, these findings indicate that while DCAF15 KO K562

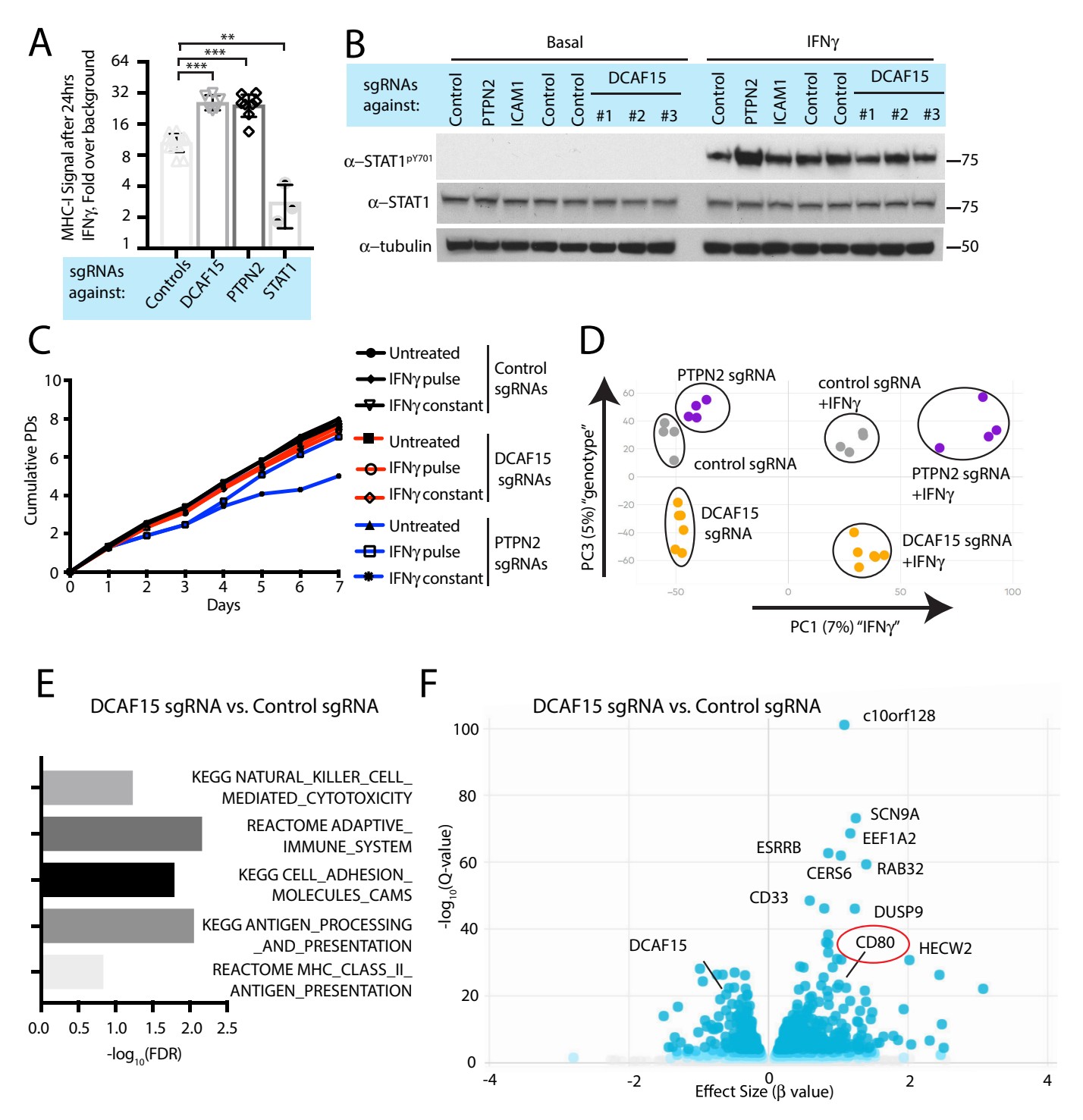

**Figure 4.** DCAF15 disruption leads to an inflamed state distinct from dysregulated IFNγ-JAK-STAT signaling. (**A**) Flow cytometry measurement of cell surface MHC-I expression in K562 cells transduced with the indicated sgRNAs after 24 hr of 10 ng/ml IFNγ treatment. ***p<0.0001; **p=0.0044, Mann-Whitney test. (**B**) Western blots illustrating total and phosphorylated STAT1 after 30 min of 1 ng/ml IFNγ treatment in K562 cells infected with the indicated sgRNAs. (**C**) Growth rate of K562 cells expressing the indicated sgRNAs, cultured under basal conditions ('untreated'), treated 24 hr with 10 ng/ml IFNγ ('IFNγ pulse'), or continuously retreated with 10 ng/ml IFNγ every day ('IFNγ constant'). (**D**) Principle component analysis (for principle components (PC) 1 and 3) of transcriptomes from K562 cells expressing the indicated sgRNAs and treated + / - 10 ng/ml IFNγ for 24 hr. (**E**) Selected GO terms, identified by RNA-seq, enriched in DCAF15 KO cells. Negative log10 transformation of the Benjamini-Hochberg corrected P value. (**F**) Volcano

*Figure 4 continued on next page*

*Figure 4 continued*

plot of genes differentially expressed between DCAF15 KO cells compared to control KO cells. Selected genes are highlighted. The FDR-corrected P-value generated from a likelihood ratio test (Q-value) is plotted against an approximate measure of the fold change in expression (Beta value).

DOI: https://doi.org/10.7554/eLife.47362.011

The following figure supplement is available for figure 4:

**Figure supplement 1.** Transcriptional responses of K562 DCAF15 KO and PTPN2 KO cells to IFNγ.
DOI: https://doi.org/10.7554/eLife.47362.012

cells exhibit a relatively normal response to IFNγ stimulation, they are nonetheless in an inflamed state primed to interact with cytotoxic lymphocytes.

## DCAF15 knockout cells enhance NK-92 triggering via CD80 expression

Intriguingly, differential expression analysis showed that one of the most significantly upregulated genes in DCAF15 KO K562 cells was CD80 (*Figure 4F*; Q value 3.7e-23, Beta value 0.98). CD80 is an important co-stimulatory molecule for lymphocytes, regulating T cell activation and tolerance by ligation to CD28, CTLA4 or PDL1 (*Chen and Flies, 2013*). During antigen-presenting cell (APC) activation, the upregulation of MHC molecules and CD80 provide critical antigenic and costimulatory signals to T cells (*Acuto and Michel, 2003*). K562 cells are an undifferentiated and multipotential CML cell line, well-studied for their ability to differentiate towards many different lineages, including APC-like states (*Lindner et al., 2003*). We hypothesized that upregulation of MHC-I and CD80 in DCAF15 KO cells may reflect a broader acquisition of APC-like properties. Indeed, immunophenotyping of unstimulated DCAF15 KO cells revealed higher levels of the APC markers CD80, CD40 as well as class I and II MHC molecules (*Figure 5A*; 2.23-fold CD80 increase; 2.01-fold CD40 increase; 1.44-fold MHC-I increase; 1.22-fold MHC-II increase). DCAF15 KO cells did not display higher levels of the APC maturation marker CD83 (*Figure 5A*). Expression levels of B7H6, ICAM1, ULBP2/5/6, IFNGR1, CD58 and NECTIN2 were either unaltered in DCAF15 KO cells or modestly changed in a fashion not expected to increase sensitivity to NK cells (*Figure 5—figure supplement 1A*). Importantly, the changes to the DCAF15 KO cell immunophenotype could be rescued by constitutive expression of a sgRNA-resistant DCAF15 open-reading frame (*Figure 5—figure supplement 1B–D*).

Transducing tumors with the B7 ligands CD80 or CD86 can enhance anti-tumor immunity by enabling the tumor cells to directly deliver antigenic and costimulatory signals to T and NK cells (*Townsend and Allison, 1993*; *Chen et al., 1992*; *Wilson et al., 1999*; *Galea-Lauri et al., 1999*). While best understood in the context of T cell biology, B7 ligands have been shown to promote NK activation, via CD28-dependent and -independent pathways (*Wilson et al., 1999*; *Galea-Lauri et al., 1999*; *Chambers et al., 1996*; *Azuma et al., 1992*; *Martín-Fontecha et al., 1999*). We confirmed that NK-92 cells are CD28-positive, whereas we could not detect CD28 on peripheral CD3- CD56+ NK cells (*Figure 5—figure supplement 1E–F*). K562 cells stably over-expressing wild-type CD80 were generated (*Figure 5B*; 49-fold higher CD80 levels than endogenous). Over-expression was sufficient to increase K562 sensitivity to NK-92 mediated killing (*Figure 5C*), whereas overexpression of a mutant form of CD80 carrying point mutations that abrogate CD28 binding (*Peach et al., 1995*) (CD80^{Q65A,M72A}) had no effect.

We next determined whether the increased CD80 expression in DCAF15 KO cells was important for their altered NK-92 sensitivity. Changes in NK-92 degranulation were measured after incubation with K562 cells pretreated with either control or CD80 blocking antibodies (*Figure 5D*). As expected, ICAM1 KO cells triggered less NK-92 degranulation than control cells (by 52 ± 12%) and were not significantly affected by CD80 antagonism (*Figure 5E*). DCAF15 KO cells showed a 17 ± 8% increased ability to trigger NK-92 cells and were approximately twice as sensitive to CD80 antagonism compared to control cells (*Figure 5E–F*). Following CD80 antagonism, degranulation triggered by DCAF15 KO was not significantly different from untreated control cells. Taken together, these results indicate that DCAF15 disruption in K562 cells induces an APC-like immunophenotype conducive to promoting lymphocyte responses, with higher CD80 expression especially important for increased NK-92 cell triggering.

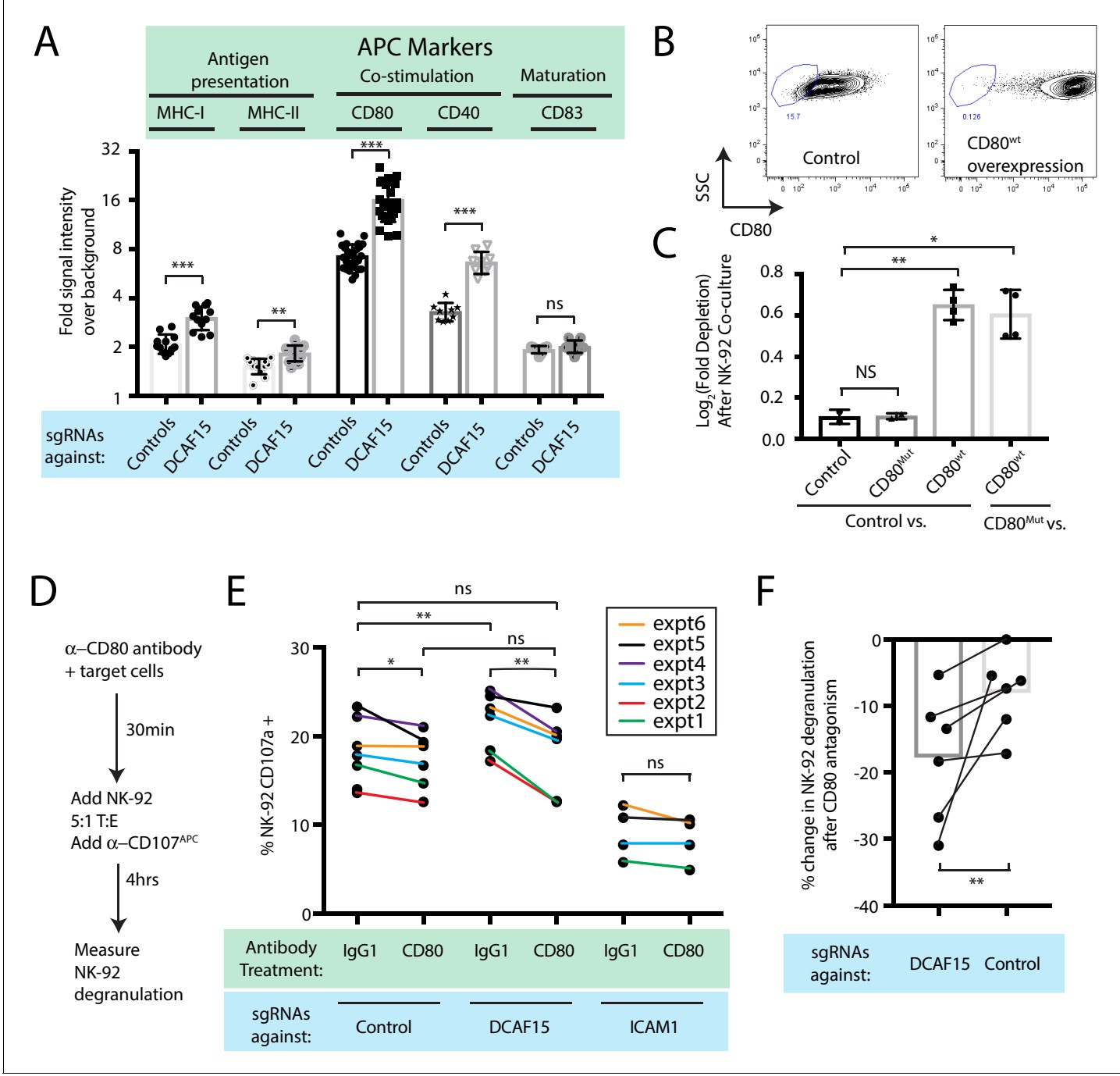

**Figure 5.** DCAF15 knockout cells enhance NK-92 triggering via CD80 expression. (**A**) Flow cytometry measurements of the indicated cell surface markers in K562 cells expressing the indicated sgRNAs. N = 9–24 samples per condition. ***p<0.0001, **p=0.001, ns p>0.1, Mann-Whitney Test. (**B**) Flow cytometry measurement of CD80 surface expression in control K562 cells or those transduced with lentivirus to overexpress CD80. Gate shows level of background fluorescence in unstained cells. (**C**) Results of competitive co-culture between indicated K562 cell types and NK-92 cells, performed at 1:1 E:T ratio. K562 cells were unmanipulated ('control') or overexpressed wild-type CD80 (CD80$^{wt}$) or mutant CD80 (CD80$^{mut}$; contains Q65A and M72A point mutations that abrogate CD28 binding). *p=0.005, **p=0.0007, NS p>0.1, unpaired T test. (**D**) xperimental design of CD80 blockade experiment. (**E**) Effect of blocking antibodies to CD80 on NK-92 activation, measured by CD107 flow cytometry on NK-92 cells after 4 hr of co-culture. Data points from experiments performed on the same day are joined by lines of the same color. **p=0.03, *p=0.06, ns p>0.1, Wilcoxon matched-pairs signed rank test. Experiment performed four times, 2x sgRNAs per condition. (**F**) Percent decrease in NK-92 degranulation after CD80 antibody treatment of indicated target cells. Data points from experiments performed same day are joined by lines. Mean is indicated. **p=0.03, Wilcoxon matched-pairs signed rank test.

*Figure 5 continued on next page*

*Figure 5 continued*

DOI: https://doi.org/10.7554/eLife.47362.013

The following figure supplement is available for figure 5:

**Figure supplement 1.** Additional immunophenotypic characterization of K562 DCAF15 KO cells.

DOI: https://doi.org/10.7554/eLife.47362.014

## The anti-leukemia drug indisulam inhibits DCAF15 function

Aryl sulfonamide drugs have demonstrated promising anti-cancer properties in hematological malignancies (*Assi et al., 2018*). Recently, it was discovered that these agents work by binding DCAF15 and redirecting the ubiquitination activity of the CRL4-DCAF15 E3 ligase towards the essential splicing factor RBM39 (*Han et al., 2017*; *Uehara et al., 2017*). This mechanism of action is conceptually similar to that of the 'IMiD' thalidomide analogs, which promote the degradation of various lymphocyte transcription factors by engaging the CRL4-cereblon E3 ubiquitin ligase (*Ito et al., 2010*; *Fischer et al., 2014*; *Krönke et al., 2014*). Presumably, sulfonamides and IMiDs also impair the degradation of the normal client proteins when they induce neomorphic activity of the substrate adaptor. This has not been proven, however, as it is difficult to systematically determine the normal substrate repertoire of adaptor proteins.

We hypothesized that treating cells with low concentrations of the aryl sulfonamide indisulam would phenocopy DCAF15 depletion (*Figure 6A*). Three-day dose-response experiments revealed that K562 cells were sensitive to indisulam, and that DCAF15 disruption reduced this sensitivity, consistent with previous reports (*Uehara et al., 2017*) (*Figure 6B*). Dose-response experiments across a panel of 16 hematological cancer cell lines confirmed the reported positive relationship between DCAF15 mRNA expression levels and indisulam sensitivity (*Han et al., 2017*) (*Figure 6—figure supplement 1A*; $R^2 = 0.33$, p=0.02).

We empirically determined that 100 nM indisulam treatment moderately lowered RBM39 levels while minimally affecting cell viability and proliferation over a four-day period (Figure 6C-D and *Uehara et al., 2017*). Remarkably, this treatment regime was able to recapitulate the increased CD80 expression seen in K562 DCAF15 KO cells (*Figure 6E*; 2.14-fold increase), and more modestly, the effects on MHC-I and CD40 expression (1.36-fold and 1.27-fold increase, respectively). CD80 upregulation was first detected 24 hr after treatment initiation and plateaued after 48 hr (*Figure 6—figure supplement 1B*). Importantly, indisulam treatment did not further upregulate CD80 in K562 DCAF15 KO cells, suggesting that the pharmaco-modulation of CD80 levels was entirely mediated through DCAF15 (*Figure 6F*).

To extend these observations to other cell lines, a panel of hematological cancer cell lines was screened to identify those with detectable CD80 expression (*Figure 6—figure supplement 1C*). The CML cell line KU812 expressed similar levels of CD80 as K562, whereas the Daudi lymphoma cell line expressed significantly higher basal CD80 levels. These cells lines were subjected to similar 4 day low-dose regimes of indisulam, which only modestly affected the growth and viability of the cells (*Figure 6—figure supplement 1D*). Both Daudi and KU812 cells up-regulated CD80 levels after indisulam treatment (*Figure 6G*; 2.45-fold for Daudi, 1.71-fold for KU812). Indisulam was not able to induce de novo CD80 expression in CD80-negative cell lines (*Figure 6—figure supplement 1E*). Thus, in certain cellular contexts, aryl sulfonamides are immuno-modulatory agents that alter co-stimulatory protein levels by disrupting the normal functions of DCAF15.

## Reduced DCAF15 expression is associated with improved survival in AML patients

Given the in vitro findings, we hypothesized that lower DCAF15 expression in myeloid malignancies could be associated with better clinical outcomes. We tested this hypothesis using publicly available acute myeloid leukemia (AML) datasets (*Bolouri et al., 2018*; *Ley et al., 2013*). In both adult and pediatric AML, lower expression of DCAF15 mRNA was associated with increased median overall survival time (*Figure 6H–I*). The improved survival of DCAF15-low patients was not driven by a correlation between DCAF15 expression and more aggressive AML subtypes (*Figure 6—figure supplement 1F–G*). Taken together, these findings indicate that lower DCAF15 function, achieved

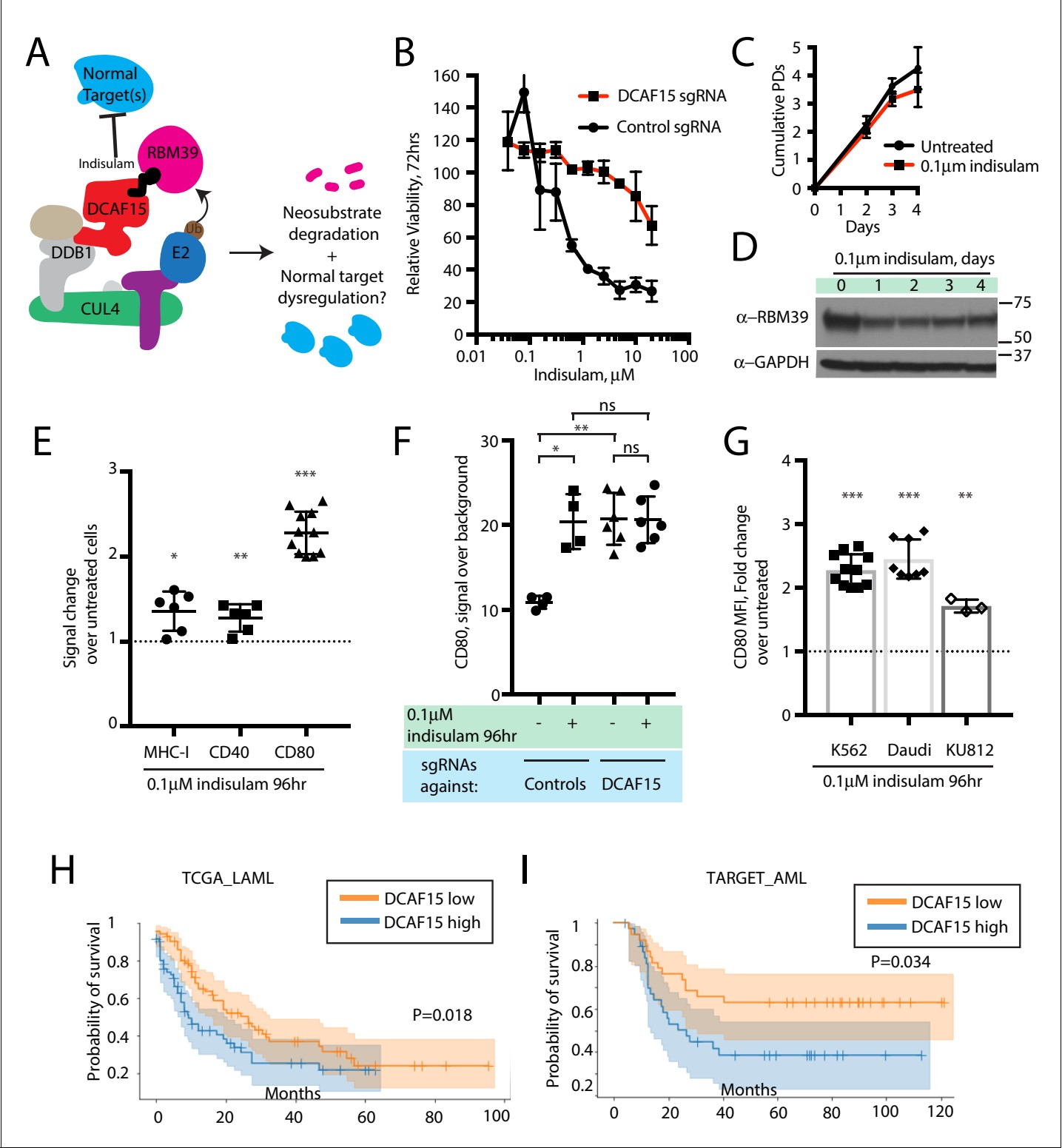

**Figure 6.** The anti-leukemia drug indisulam inhibits DCAF15 function. (**A**) Proposed model for DCAF15 gain- and loss-of-function phenotypes triggered by indisulam treatment. (**B**) Dose response of K562 cells expressing the indicated sgRNAs to indisulam. Relative viability measured by ATP content. N = 2 experiments. Mean and standard deviation are shown. (**C**) Growth of K562 cells treated with 0.1 μM indisulam over 4 days. N = 12 per timepoint. Mean and standard deviation are shown. (**D**) Western blot for total RBM39 levels in K562 cells after 0.1 μM indisulam treatment for the indicated number of days. (**E**) Flow cytometry measurements of the indicated cell surface markers in K562 cells after indisulam treatment. Mean and standard deviation are shown. ***p<0.0001, **p=0.0085, *p=0.013, mean significantly different from 1, one sample T test. (**F**) CD80 expression measured by flow

*Figure 6 continued on next page*

*Figure 6 continued*

cytometry in indisulam-treated K562 cells expressing the indicated sgRNAs. Mean and standard deviation are shown. N = 4–6 samples. **p=0.0095, *p=0.0286, ns p>0.1, Mann-Whitney test. (G) Effect of indisulam treatment on CD80 expression in the indicated cell lines. N = 3–12 samples. Mean and standard deviation are shown. ***p<0.0001, **p=0.0066, mean significantly different from 1, one sample T test. (H) Kaplan-Meier analysis of overall survival in adult AML patients from TCGA LAML project stratified by DCAF15 expression. 'DCAF15 high' and 'DCAF15 low' represents patients in top or bottom 50% of DCAF15 expression, respectively. N = 142 patients. 95% confidence interval shown. Median survival, 16.17 vs 12.18 months. P-value from log-rank test. (I) Kaplan-Meier analysis of overall survival in pediatric AML patients from TARGET project stratified by DCAF15 expression. 'DCAF15 high' and 'DCAF15 low' represents patients in top or bottom 20% of DCAF15 expression, respectively. N = 76 patients. 95% confidence interval shown. Median survival, 21.17 vs NA. P-value from log-rank test.
DOI: https://doi.org/10.7554/eLife.47362.015

The following figure supplement is available for figure 6:

**Figure supplement 1.** Additional indisulam experiments.
DOI: https://doi.org/10.7554/eLife.47362.016

pharmacologically or by genetic means, is associated with favorable immunophenotypes in vitro and improved outcomes in AML patients.

## Cohesin complex members are CRL4-DCAF15 E3 ligase client proteins

The normal substrate repertoire of DCAF15 is unknown. To systemically identify direct DCAF15 client proteins, we undertook proximity-based proteomic analysis of DCAF15 interaction partners (*Figure 7A*). DCAF15 was fused to a promiscuous bacterial biotin ligase (*Roux et al., 2012*) ('DCAF15-BioID') and stably expressed in K562 cells, enabling recovery of interaction partners by streptavidin pull-down. We first confirmed that exogenous C-terminally tagged DCAF15 was able to rescue DCAF15 KO phenotypes and associate with CRL4 complex members DDB1 and CUL4A (*Figure 5—figure supplement 1B–D* and *Figure 7—figure supplement 1A*). During stable DCAF15 overexpression, we observed that the basal concentration of biotin in the media (~3 µM) was sufficient to induce BioID activity in the absence of exogenous (50 µM) biotin supplementation (*Figure 7B*). However, proteasome inhibition by MG132 increased accumulation of DCAF15-BioID and biotinylated species. As a control, results were compared to a GFP-BioID fusion, expected to generically biotinylate proteins. GFP-BioID accumulated much more readily than DCAF15-BioID, and its biotinylation activity was not affected by MG132 treatment.

After 24 hr of biotin and MG132 treatment, biotinylated protein species were recovered under stringent denaturing conditions. Isobaric labeling and mass spectrometry were used to quantitatively compare the DCAF15 interactome to the GFP interactome (*Figure 7C* and *Supplementary file 5*). This approach clearly recovered DCAF15 and the core CRL4 complex, including DDA1, DDB1 and CUL4A (DCAF15: 155.6-fold change, p=2.5e-152; DDA1: 24.5-fold change, p=6.5e-139; DDB1: 4.86-fold change, p=8.1e-21; CUL4A: 3.8-fold change, p=8.5e-18).

Surprisingly, two of the most differentially biotinylated proteins were the cohesin complex members SMC1A and SMC3 (SMC1: 2.39-fold change, p=0.00098; SMC3: 2.76-fold change, p=5e-5). We confirmed the interaction between DCAF15-BioID and endogenous SMC1 and 3 by streptavidin-pulldown followed by western blotting (*Figure 7D*). To determine whether this association with cohesin was a generic feature of CRL4 complexes or specific to the CRL4 loaded with DCAF15, we examined the interaction partners of a different substrate adaptor. DCAF16 is a nuclear-localized CUL4 substrate adaptor, which, like DCAF15, interacts with DDB1 despite lacking a canonical WD40 docking domain (*Jin et al., 2006*). When stably expressed in K562 cells, DCAF16-BioID fusions accumulated similarly to DCAF15-BioID, interacted with DDB1 but did not biotinylate SMC proteins (*Figure 7E*). As low concentrations of indisulam phenocopy certain aspects of DCAF15 depletion (*Figure 6*), we asked whether indisulam treatment would alter the interaction of DCAF15 with cohesin. DCAF15-BioID cells were pre-treated with indisulam for 72 hr prior to biotin and MG132 addition. This treatment regime lead to substantial indisulam-dependent biotinylation of RBM39 and reduced recovery of biotinylated SMC proteins (*Figure 7E*).

To test whether DCAF15 promoted SMC1 ubiquitination, we co-transfected 293 T cells with his-tagged ubiquitin, DCAF15 and SMC1A, and purified ubiquitinated species under denaturing conditions by nickel affinity chromatography. Co-expression of DCAF15, but not GFP, led to the recovery of poly-ubiquitinated SMC1A (*Figure 7F*). Treatment with indisulam reduced the amount of

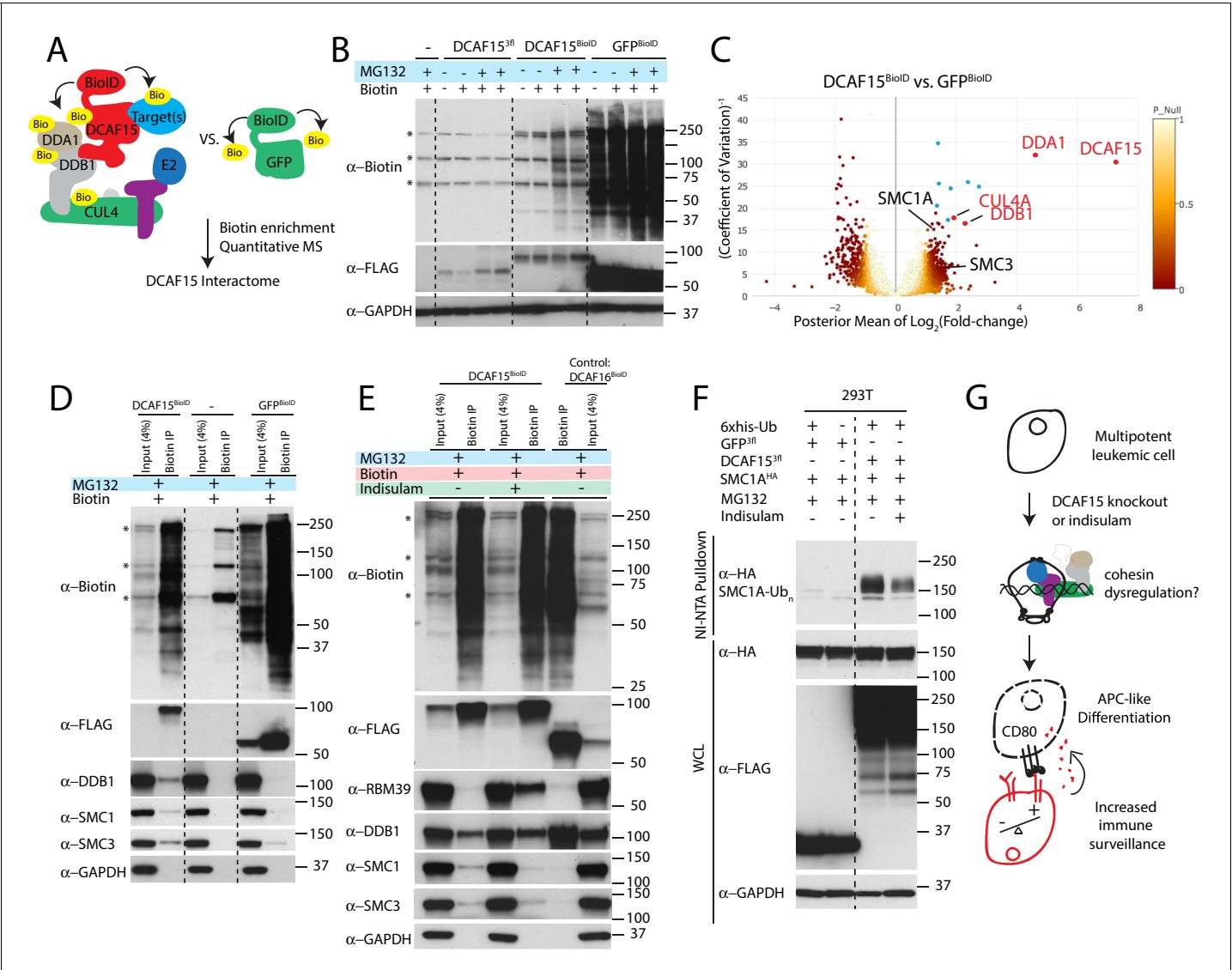

**Figure 7.** Cohesin complex members are CRL4-DCAF15 E3 ligase client proteins. (A) Experimental system for discovering DCAF15 interaction partners by proximity ligation. (B) Indicated constructs were stably expressed in K562 cells, and biotinylated proteins detected by HRP-conjugated streptavidin ('α-biotin'). Asterisks denote major endogenous biotin-containing proteins. 'MG132' and 'Biotin' refer to 18 hr treatment with 5 µM MG132 or 50 µM biotin. (C) Proteins identified by quantitative mass spectrometry as differentially biotinylated by DCAF15-BioID as compared to GFP-BioID. Log2-fold changes are plotted against precision of the measurement (1/coefficient of variation). Colors denote the posterior probability that a protein fold change was small (referred to as 'P-null' in the legend), as explained in the Materials and methods. Data points in red are select CRL4 core complex members. Data points in blue are endogenous biotin-containing proteins. (D-E) Affinity capture of biotinylated proteins by streptavidin beads ('biotin IP') and detection by western blot. Indicated constructs were stably expressed in K562 cells. 'Indisulam' refers to 48 hr 0.1 µM indisulam treatment prior to MG132 and biotin addition. (F) Capture of 6xhis-ubiquitinated species by nickel chromatography under denaturing conditions and detection by western blot. The indicated expression plasmids were transiently transfected into 293 T cells. Input samples were prepared from whole cell lysates (WCL). 'MG132' and 'Indisulam' refer to 12 hr treatment with 10 µM MG132 or 2 µM Indisulam prior to harvest. G)Model of DCAF15 function.

DOI: https://doi.org/10.7554/eLife.47362.017

The following figure supplement is available for figure 7:

**Figure supplement 1.** Functionality of exogenous DCAF15.

DOI: https://doi.org/10.7554/eLife.47362.018

ubiquitinated SMC1A recovered. These orthogonal proteomic and biochemical assays support the notion that cohesin proteins are *bona fide* client ubiquitination substrates for DCAF15, with these interactions impaired by DCAF15-engaging aryl sulfonamide drugs.

## Discussion

Clinical and experimental data have revealed that disrupting optimal antigen presentation levels is a common mechanism by which cancer cells escape recognition by the adaptive immune system. We performed CRISPR-Cas9 screens using NK-92:K562 co-cultures to uncover perturbations that enhance natural killer mediated anti-cancer immunity. We discovered and characterized how disruption of DCAF15 or PTPN2 sensitizes a variety of cancer cell types to both NK-92 and primary NK cells. In addition, the screens clearly identified known lymphocyte adhesion factors and aNKR/iNKR ligands, such as ICAM1, NCR3LG1, CD58, CD84 and NECTIN2. Performing additional genetic screens on diverse cancer cell lines and natural killer subtypes will enable a more complete understanding of the relative importance of various aNKRs and iNKRs and should identify novel immunotherapeutic targets.

There was a strong IFNγ signature within our NK-92:K562 screens. We determined that disruption of PTPN2, a negative regulator of IFNγ signaling, consistently enhanced NK cell sensitivity. Interestingly, preventing cancer cell IFNγ signaling had a much more variable effect than removing negative feedback on the pathway, promoting resistance or sensitization to NK cells in target and effector cell dependent manner. This variability likely reflects the complexity of cancer cell IFNγ signaling, which can include MHC upregulation, growth suppression and immunomodulation.

In K562 cells, PTPN2 KO rendered IFNγ treatment growth suppressive, likely through the induction of apoptosis, while also enhancing the immunomodulatory effects of the cytokine. The immunophenotypic changes include enhanced MHC-I expression, which likely inhibits full activation of NK cell cytotoxicity, especially in primary NK cells that express a broader KIR repertoire than NK-92 cells (*Maki et al., 2001*). PTPN2 disruption has previously been shown to enhance activated T-cell mediated killing, as well as potentiate the effect of immunotherapy in syngeneic tumor models (*Pan et al., 2018*; *Manguso et al., 2017*; *Luo et al., 2018*). These data suggest that targeting PTPN2 may be a generalizable strategy to sensitize tumor cells to multiple arms of the immune system.

The importance of the cancer cell IFNγ response prompted us to systemically identify perturbations that modulated IFNγ signaling, as read out by cytokine-induced MHC-I upregulation. As expected, this screen clearly delineated the proximal components of the IFNγ-JAK-STAT pathway, as well as the antigen processing/presentation machinery. Surprisingly, disruption of DCAF15, a poorly characterized substrate adaptor for the CRL4 E3 ligase, was a top scoring hit in both the MHC-I and NK screens. We therefore focused our efforts on understanding the role of DCAF15 in this context. We determined that DCAF15 KO cells did not have a grossly dysregulated response to IFNγ, but scored in the MHC-I screen due to higher unstimulated levels of MHC-I. This change reflected a broader phenotypic switch in DCAF15 KO cells reminiscent of APC activation, including the upregulation of a variety of co-stimulatory and antigen-presenting molecules (*Figure 7G*). Higher CD80 levels in DCAF15 KO cells were especially important for increasing NK-92 cell triggering. Cross-talk between APCs and NK cells is a well-established phenomenon that mutually regulates both cell types (*Ferlazzo and Münz, 2004*). These interactions, which often occur at sites of inflammation or secondary lymphoid organs, can promote APC maturation or lysis in a context-dependent fashion. Further work is needed to determine whether DCAF15 plays a role in the activation of APCs and their interactions with NK cells.

Our data suggest that the surface factors upregulated by DCAF15 disruption are not direct client proteins of the substrate receptor, but rather represent events secondary to altered turnover of the normal DCAF15 client protein(s). We pioneered a novel approach to systemically purify substrates of DCAF family members, as conventional biochemical methods are poorly suited to recover the transient, low-affinity interactions between substrate adaptors and client proteins. The use of DCAF-BioID fusion proteins and proteasome inhibition protects labile substrates from degradation and robustly recovers biotinylated proteins under stringent, denaturing conditions. We anticipate this will be a generalizable strategy for discovering client proteins for the whole family of CUL4 CRL substrate receptors.

Proteomic analysis and subsequent validation experiments showed that DCAF15 loaded into CLR4 complexes, interacted with cohesin complex members SMC1 and SMC3, and promoted their ubiquitination. We were intrigued by these interactions given the similar CRISPR screening scoring pattern of DCAF15 to the cohesin factors STAG2 and HDAC8 (*Figure 2E*); the shared roles of cohesin and CRL4 E3 ligases in DNA metabolism, organization, replication and repair (*Uhlmann, 2016*; *O'Connell and Harper, 2007*; *Litwin et al., 2018*); and the ability of cohesin mutations to dysregulate hematopoietic differentiation in myeloid malignancies (*Mazumdar and Majeti, 2017*). Rather than globally controlling SMC protein levels, we speculate that CRL4-DCAF15 complexes ubiquitinate cohesin at specific genomic sites to regulate chromatin topology or repair (*Figure 7G*). In this model, disruption of either STAG2, HDAC8 or DCAF15 impair cohesin function with overlapping phenotypic consequences. Further work is needed to elaborate the control of cohesin function by DCAF15 and how this may promote APC-like differentiation.

Recently, it was discovered that aryl sulfonamide drugs including indisulam are capable of binding to DCAF15 and altering CRL-DCAF15 substrate specificity towards the splicing factor RBM39 (*Han et al., 2017*; *Uehara et al., 2017*). The cytotoxic effects of indisulam were driven by splicing defects resulting from RBM39 degradation. Our studies confirmed indisulam-induced RBM39 degradation and the indisulam-dependent interaction between DCAF15 and RBM39. We also discovered indisulam-induced phenotypes attributable to inhibition of DCAF15's normal functions. Concentrations of indisulam with limited RBM39 degradation or cytotoxicity had immunomodulatory properties that phenocopied genetic DCAF15 disruption. Biochemically, the recruitment of endogenous client proteins to CRL4-DCAF15 and subsequent ubiquitination was impaired by indisulam treatment. We also determined that AML patients with naturally occurring lower levels of DCAF15 had improved overall survival. While these clinical data are preliminary in nature, they provide a rationale for drugging DCAF15 in myeloid neoplasms, achieved through judicious dosing of existing anti-cancer sulfonamides or the development of pure DCAF15 inhibitors.

# Materials and methods

## Cell lines

All cell lines were purchased from ATCC and were tested monthly for mycoplasma contamination. Cell lines other than NK-92 were maintained in RPMI supplemented with 10% FBS, 1 mM GlutaMAX and 1% antibiotic, antimycotic. NK-92 cells were grown in Myelocult H5100 (Stem cell Technologies) supplemented with 100 U/ml human IL-2 (Peprotech cat#200–02). IL-2 stock solution was made by reconstituting lyophilized cytokine to 10e6 U/ml in 50 mM acetic acid, 0.1% BSA in PBS.

## Construction of the CRISPR Library

CRISPR screening was performed using bespoke genome-scale libraries, to be described in detail elsewhere. In brief, the sgRNA library was designed with 120,021 sgRNAs present, representing six guides each against 21,598 genes and four guides each against 1918 miRNAs, as well as 1000 non-targeting negative control guides (*Supplementary file 1*). sgRNAs targeting protein-coding genes were based on the Avana libraries (*Doench et al., 2016*). sgRNAs targeting miRNAs were based on the design of the Gecko v2.0 libraries (*Sanjana et al., 2014*). The protospacer library was synthesized by Twist Biosciences. The synthesized oligo library was amplified using emulsion PCR followed by purification. The vaccinia virus DNA polymerase was used to clone the protospacers into a lentiviral construct (In-fusion, Takara). The lentiviral construct was linearized by Bfua1 digestion (New England Biolabs). To ensure complete digestion, Bfua1 activity was stimulated by addition of 500 nM of a double-stranded oligo containing a Bfua1 site (5' atagcacctgctata 3') (based on *Grigaite et al., 2002*).

The guide RNAs were expressed from a human U6 promoter, using a modified sgRNA design (A-U flip, longer stem-loop) previously described (*Chen et al., 2013*). An EF1a-Puro-T2A-cerulean-wpre ORF was used for selection purposes and for measuring infection rates. The library was electroporated into MegaX cells and plated across $37 \times 500$ cm$^2$ LB-carbenicillin plates. The library was recovered and pooled by scraping and column-based plasmid purification (Zymopure GigaPrep).

## Preparation of virions

16M 293 T cells were plated onto 177 cm$^2$ dishes (9x plates total). 24 hr later, cell media was replaced with 32mls of DMEM+10% FBS (D-10). Cells were transfected with the library by lipofection (per plate: 158 µl lipofectamine 2000, 8mls of Optimem, 3.95 µg of VSVG, 11.8 µg of Pax2, 15.78 µg of library). The transfection mixture was left on the cells overnight, then changed to 25mls of D-10. Viral supernatant was collected 48 hr later. Debris was removed by centrifugation at 200 g. Aliquots were flash-frozen and stored at −80℃.

## Validation of the Cas9-expressing cell line used for screening

K562 cells were lentivirally infected with an EF1a-spCas9-T2A-blastR construct. Following 10 µg/ml blasticidin selection, cells were dilution cloned. Clonal isolates with high levels of cas9 expression (as determined by western blot) were selected for further characterization. To determine the kinetics and efficacy of gene cutting under screening conditions (e.g., low multiplicity of infection (MOI)) of the sgRNA construct, cells were infected with a lentiviral construct expressing both EGFP and a sgRNA targeting GFP. Following puromycin selection, the loss of EGFP expression was monitored by flow cytometry. In other experiments, the ability to effectively deplete endogenous proteins was determined by using a series of sgRNAs targeting the mismatch repair complex and measuring protein depletion by western blot.

## NK screen

Five hundred million spCas9-expressing K562 cells were mixed with CRISPR sgRNA library virions and 1L of media, then distributed across 34 6-well plates. Cells were spin-infected at 1900 rpm for 30 min at room-temperature with CRISPR library virus, conditions calculated to achieve a MOI of ~0.3 and 1000 cells per sgRNA library representation. MOI was measured by tracking the percent of the population expressing the cerulean marker found in the sgRNA library. Cells were incubated overnight in viral supernatant prior to being pooled, spun down to remove virions, and returned to spinner-flask culture (Bell-Flo Flask, Bellco Glass Inc). 24 to 48 hr post-infection, 2 µg/ml puromycin selection was started for four days. Cells were maintained in log-growth phase with a minimal representation of 500 M cells. Seven days post guide-infection, NK cell challenge was initiated. For the 2.5:1 E:T challenge, 100M K562 cells were mixed with 250M NK-92 cells in 1L of Myelocult, and split across 25 177 cm$^2$ dishes. For the 1:1 E:T challenge, 100M K562 cells were mixed with 100M NK-92 cells in 400mls of Myelocult, and split across 10 177 cm$^2$ dishes. As a control, K562 cells were continuously propagated in RPMI media. 2 days after initiating the NK-92 challenge, the co-culture was switched to RPMI media. NK-92 cells, when grown in RPMI without IL-2, are rapidly lost from the culture. Six days after the media switch, 100 M cell pellets were generated for library construction. All cell pellets were stored cryopreserved in cell-freezing media (Gibco #12648).

## MHC-I screen

Generation of K562 cells expressing the sgRNA library was performed as described above. On day 8 post-guide infection, 150 M cells were stimulated with 10 ng/ml IFNγ (R and D Systems 285-IF). 24 hr later, cells were spun down, washed, and stained in 3mls of 1:200 anti-HLA ABC-APC (W6/32 clone, BioLegend) in PBS, 2% FBS for 30 min at 4℃. Approximately, 10M of the brightest 20% and dimmest 20% of cells were sorted (BD FACSAria Fusion). Cell purity was determined to be >95% by re-analysis post-sorting. A replicate of this experiment was performed on day 14 post-guide infection.

## CRISPR library preparation and sequencing

Genomic DNA was prepared by thawing cryopreserved cell pellets and proceeding with DNA extraction using column-based purification methods (NucleoSpin Blood XL, Machery-Nagel). Protospacer libraries were generated by a two-step PCR strategy, modified from *van Overbeek et al. (2016)*. In the first round of PCR, 150 µg of gDNA (equivalent to 15M K562 cells and 125-fold coverage of the library) was used in a 7.5 ml PCR reaction to amplify the protospacers. This reaction was performed using a 500 nM mixture of primers containing 0–9 bp staggers, to ensure base-pair diversity during Illumina sequencing (see *Supplementary file 7*). The reaction was performed with Phusion master mix (New England Biolabs) and 3% DMSO with the following cycling conditions: one

cycle X 30 s at 98°C, 21 cycles X 15 s at 98°C, 20 s at 63°C, 15 s at 72°C, one cycle X 2 min at 72°C. In the second round of PCR, 4 μl of the initial PCR product was used as the template in a 200 μl PCR reaction to make the sample compatible with Illumina chemistry and to add unique I5 and I7 barcodes to the sample. The reaction was performed with Phusion master mix (New England Biolabs), 500 nM primers and 3% DMSO with the following cycling conditions: one cycle X 30 s at 98°C, 12 cycles X 15 s 98°C, 20 s at 60°C, 15 s at 72°C, one cycle X 2 min at 72°C. The library was size-selected first by a 1:1 SPRI bead selection (AMPure XP beads, Beckman Coulter), quantified by high-sensitivity dsDNA Qubit (ThermoFisher Scientific), and pooled. An agarose size selection step (PippinHT, Sage Science) was performed prior to sequencing on an Illumina Hiseq4000.

## Screen analysis

Libraries were sequenced to a depth of ~20 million reads per condition. Reads were aligned to the sgRNA library using bowtie2. Protospacer count tables were generated from these alignments with python scripts and processed with MAGeCK. MAGeCK analysis was used to score and prioritize sgRNAs, using default settings in the algorithm (*Li et al., 2014*). A subset of genes, mostly from highly related gene families, have more than six sgRNAs targeting them. As the MAGeCK scoring method tends to prioritize consistency of effect over magnitude of effect, genes with more than six guides targeting them were excluded from the analysis. MAGeCK scores were -log10 normalized, and values were plotted against FDR values.

## NK-92 competitive co-culture assay

Cas9-expressing K562 or Daudi cells were transduced at a high MOI with a lentiviral sgRNA construct expressing the guide of interest, puromycin resistance and CMV promoter-driven expression of either a red or green fluorescent protein. Knockout cell populations were maintained in a polyclonal state. Gene disruption was confirmed at the protein level by flow cytometry or western blot, and by RNA-seq when antibody reagents were not available. Complete knockout in >90% of the population was routinely achieved (*Figure 2—figure supplement 1* and *Figure 3—figure supplement 1*). Cells were counted and 0.25M red-labeled test cells were mixed with 0.25M green-labeled control cells (expressing a sgRNA against an olfactory receptor gene). The mixture was either grown in RPMI or mixed with 1.25M NK-92 cells in 4mls Myelocult media in 6-well plates. The ratio of green-to-red cells was measured 2 to 4 days post-challenge, with the fold-change normalized to the ratio in the non-challenged state (to control for differences in basal cell growth rate).

## Primary NK cell isolation

PBMCs were isolated from leukocyte-enriched blood of human donors (Stanford Blood Center). Donors were not genotyped or pre-screened for infectious disease markers. Natural killer cells were isolated by negative selection using magnetic columns (Miltenyi 130-092-657). Purity post-selection was routinely >97% CD56+ CD3-. Isolated NK cells were plated at 1 M/ml density in 96-well u-bottom plates and stimulated overnight with 1000 U/ml IL-2 in complete RPMI with 10% FBS.

## Primary NK cell degranulation assay

Following IL-2 activation, NK cells were counted and mixed with target cells in 96-well u-bottom plates. 100,000 NK cells were mixed with 40,000 target cells in a final volume of 200 μl for a 2.5:1 E:T ratio. Cells were co-cultured at 37°C for 2 hr, then stained with anti-CD56-BV421 (1:200) and anti-CD107a-APC (1:200 dilution) for 30 m at 4°C. CD107a expression was assayed by flow cytometry in the 7-AAD- CD56+ RFP- GFP- cell population. Primary NK cells exhibited very little basal degranulation in the absence of target cells (<1% CD107a+). The assay was repeated with six primary donors with technical duplicates.

## Primary NK cell competitive co-culture assay

Following IL-2 activation, NK cells were counted and mixed at a 1:1 E:T ratio with RFP-labeled target cells and GFP-labeled control cells in a 96-well u-bottom plate. 50,000 NK cells were combined with 25,000 Red and 25,000 Green cells in a final volume of 200 μl and continuously expanded in 96-well plates to as needed. The ratio of GFP+ to RFP+ cells was measured by flow cytometry on days 2 and 4 post-challenge, and the fold change was normalized to the ratio in the non-challenged state

(to control for differences in basal cell growth rate). The assay was repeated with six primary donors with technical duplicates.

## Flow cytometry

0.5 to 1 M cells were spun down and stained with APC-conjugated antibodies for 30mins at 4°C in PBS with 2% FBS. Cells were analyzed on a BD LSRFortessa. The geometric mean fluorescence intensity of singlet, DAPI-excluding cells was measured, and normalized to the background fluorescence of that particular genotype of cells.

## DCAF15 rescue experiments

The DCAF15 open reading frame was synthesized based off reference sequence NM_138353, but with silent mutations designed to confer resistance to all three sgRNAs. To ensure resistance to sgRNA-directed gene cutting, silent mutations were introduced to the PAM domain and the proximal region of the protospacer (guide #1: 5' GCTGCACACCAAGTACCAGGTGG to GCTGCACAC-CAAaTAtCAaGTaG. Guide #2: 5' TGACATCTACGTCAGCACCGTGG to TGACATCTACGTCtcCAC aGTaG; Guide #3: 3' GCAGCTTCCGGAAGAGGCGAGGG to GCAGCTTCCGGAAtaaaCGtGGt).

The rescue construct was expressed lentivirally from an EF1a promoter, with a c-terminal 3-flag epitope tag. Rescue cells were selected with hygromycin. The rescue construct was introduced 2 weeks after the initial introduction of the DCAF15 sgRNAs, and stable cell lines generated by a week of selection in 375 µg/ml hygromycin. Expression of the construct was confirmed by lysing cells in PBS+0.1% NP40 (as per *Han et al., 2017*) and western blotting for the Flag tag.

## CD80 blocking experiments

RFP-labeled target cells (0, 0.75M, or 2.5M) were resuspended in 0.5mls Myelocult + 5 µg/ml control (MOPC-21 clone) or blocking CD80 (2D10 clone) antibodies. Cells were incubated for 30 mins at room temperature. 0.5M NK-92 cells in 0.5mls Myelocult were added to the well with 1:200 anti-CD107a-APC antibody. Cells were co-cultured for 4 hr at 37°C. Flow cytometry was used to measure CD107A-APC expression in the NK-92 (RFP-negative) population. NK-92 cells were found to display a basal level of CD107a expression in the absence of effector cells. Increases above basal staining levels were used to define NK-92 degranulation. The experiment was repeated four times to establish biological replicates.

## CD80 over-expression experiments

A full-length CD80 open reading frame, based on reference sequence NM_005191.4, was synthesized with a c-terminal 3x Flag tag and expressed lentivirally from the EF1a promoter. Over-expressing cells were selected with hygromycin. As a control construct, a mutant version of CD80 was synthesized with Q65A and M72A mutations in the 'V-type' Ig-like domain. Each of these residues makes contacts with CTLA4 in published CD80-CTLA4 co-crystal structures (*Stamper et al., 2001*) and alanine scanning experiments have shown that these residues are required for CD80 binding to CD28 or CTLA4 (*Peach et al., 1995*).

## sgRNA sequences used

See *Supplementary file 6*.

## Antibodies used

See *Supplementary file 8*.

## Cell viability after in vitro cytokine stimulation

Cells were plated at 0.25 M/ml in media with or without 10 ng/ml IFNγ. Every day, an aliquot of cells was counted (Vi-CELL XR, Beckman Coulter), and cells were diluted in fresh media to maintain them in logarithmic growth phase. Continuously-treated cells were re-fed with fresh 10 ng/ml IFNγ every day, whereas pulse-treated cells only received 24 hr of cytokine treatment.

## Western blotting

Cells were counted, spun down and washed in PBS prior to lysis in NP40 lysis buffer (25 mM HEPES pH 7.5, 150 mM NaCl, 1.5 mM MgCl$_2$, 0.5% NP40, 10% glycerol, 2 mM DTT) with protease and phosphatase inhibitors. Lysates were normalized by cell count and/or protein concentration. 20–40 µg of lysates were used for western blot analysis using standard procedures. Antibody binding was detected by enhanced chemiluminescence (SuperSignal Dura, Thermo Scientific).

## RNA-sequencing analysis

Cells were seeded at 0.33 M/ml density. Twenty-four hours later, 2 M cells were collected. RNA was first purified by TRizol extraction and then further purified using column-based methods and polyA selection. RNAseq libraries were constructed using TruSeq Stranded mRNA Library Prep Kits (Illumina) and were sequenced on an Illumina Hiseq4000 machine using 150 bp paired-end reads. Transcript abundances were quantified using Salmon (v0.9.1) in pseudo-alignment mode, without adapter trimming, using the Ensembl GRCh38 transcriptome (*Patro et al., 2017*). Differential expression analysis was performed using Sleuth (v0.29.0) (*Pimentel et al., 2017*). RNA-seq analysis was executed and visualized using an in-house web-based platform. RNA sequencing data are available under accession number GEO:GSE134173.

## Indisulam treatment

Indisulam (Sigma SML1225) was reconstituted at 10 mM in DMSO and stored in single use aliquots at −80°C. For indisulam dose-response experiments, 5000 cells were plated in 384-well plates (Greiner) and treated with indisulam over a 72 hr period. A 12-point dose-response was performed between 10 µM and 4.9 µM of drug, as dispensed by a Tecan D300e. Cell viability was measured by ATP quantification (Cell Titer Glo, Promega). Dose-response measurements were fitted to a sigmoidal curve and an IC50 determined (Prism, GraphPad Software). DCAF15 expression data from different cell lines was downloaded from the Cancer Cell Line Encyclopedia (*Barretina et al., 2012*) (https://portals.broadinstitute.org/ccle).

For low-dose indisulam experiments, cells were plated at 0.4 M/ml in media with or without 100 nM indisulam (1:100,000 dilution of stock). Cells were diluted every day in fresh media and drug to maintain them in logarithmic growth phase. Cells were analyzed for CD80 expression 96 hr after initiation of treatment.

## AML survival analysis

FPKM RNAseq quantification for patient samples from the TARGET and TCGA LAML cohorts was obtained from the NIH NCI Genomic Data Commons DATA portal (https://portal.gdc.cancer.gov/). Clinical data for that TARGET AML and TCGA LAML cohorts were obtained from the Genomic Data Commons DATA portal and the Broad Institute TCGA Genome Data Analysis Center (http://gdac.broadinstitute.org/) respectively. Patients with matching clinical and transcript abundance data patients were stratified by DCAF15 expression as indicated. Survival time and vital status were defined as 'Overall Survival Time in Days' and 'Vital Status' respectively for the Target AML cohort. For the TCGA LAML study survival time for deceased patients ('patient.vital_status'=dead) was defined as 'patient.days_to_death' while for living patients ('patient.vital_status'=alive) 'patient.days_to_last_followup' was used for survival time. Survival analysis and Kaplan–Meier plots were generated using lifelines software for python (https://doi.org/10.5281/zenodo.2584900).

## DCAF15 proximity ligation

K562 cells were infected with lentivirus expressing DCAF15-3flag-BioID-HA-T2A-BlastR, DCAF16-3flag-BioID-HA-T2A-BlastR, or 3flag-GFP-T2A-BioID-HA-T2A-BlastR. Stable cell lines were generated by 10 µg/ml blasticidin selection. In triplicate, 20 M cells were grown in media supplemented with 50 µm biotin and 5 µm MG132. 18 hr later, cell pellets were washed three times with ice cold PBS and lysed in mild lysis buffer (PBS 0.1% NP40 + PI/PPI), conditions determined to maximize the solubility of over-expressed DCAF15. 1 mg of clarified lysate was used for enrichment of biotinylated species. Lysates were mixed with 60 µl of streptavidin beads (Pierce #88817) for 4 hr at 4°C in 500 µl total volume. Beads were collected on a magnetic column, washed twice with 1 ml RIPA buffer (25

mM HEPES-KOH pH 7.4, 150 mM NaCl, 1% Triton X-100, 0.5% sodium deoxycholate, 0.1% SDS, 1 mM EDTA) and three times with 1 ml urea buffer (2M urea, 10 mM TRIS-HCl pH 8.0).

For western blotting experiments, after urea washing, beads were equilibrated in mild lysis buffer. Elution was performed by incubating the beads 15 min at 23°C, 15 min at 95°C in 30 μl 2.5X Laemlli buffer supplemented with 10 mM biotin and 20 mM DTT. The eluate was brought down to 1X concentration with lysis buffer prior to western blotting. We note that detection of total biotinylated species by western blot was extremely sensitive to detection conditions. Membranes were blocked for 10 min in PBS, 2.5%BSA, 0.4% Triton-X 100. 1 ng/ml streptavidin-HRP in blocking buffer was added for 25 min at room temperature. The membrane was washed for 15 min in PBS 0.4% TritonX-100 prior to ECL exposure.

To measure BioID activity after indisulam treatment, cells were treated for 48 hr with 0.1 μm indisulam, followed by 24 hr in indisulam with 50 μm biotin and 5 μm MG132.

## Proteomic analysis of DCAF15 interaction partners

For proteomics experiments, after the urea buffer washes, the samples were resuspended in denaturing buffer (8M urea, 50 mM ammonium bicarbonate pH 7.8). Proteins were reduced with dithiothreitol (5 mM, RT, 30 min) and alkylated with iodoacetamide (15 mM RT, 45 min in the dark). Excess iodoacetamide was quenched with dithiothreitol (5 mM, room temperature, 20 min in the dark). The samples were diluted to 1 M urea using 50 mM ammonium bicarbonate and then digested with trypsin (37°C, 16 hr). After protein digestion, samples were acidified with trifluoroacetic acid to a final concentration of 0.5% and desalted using C18 StageTips (*Rappsilber et al., 2007*). Peptides were eluted with 40% acetonitrile/5% formic acid then 80% acetonitrile/5% formic acid and dried overnight under vacuum at 25°C (Labconco CentriVap Benchtop Vacuum Concentrator, Kansas City, Mo).

For tandem mass tag (TMT) labeling, dried peptides were resuspended in 50 μL 200 mM HEPES/ 30% anhydrous acetonitrile. TMT reagents (5 mg) were dissolved in anhydrous acetonitrile (250 μL) of which 10 μL was added to peptides to achieve a final acetonitrile concentration of approximately 30% (v/v). Following incubation at room temperature for 1 hr, the reaction was quenched with 5% hydroxylamine/200 mM HEPES to a final concentration of 0.3% (v/v). The TMT labeled peptides were acidified with 50 μL 1% trifluoroacetic acid and pooled prior to desalting with SepPak (Waters) and dried under vacuum.

The pooled TMT-labeled peptides were fractionated using high pH RP-HPLC. The samples were resuspended in 5% formic acid/5% acetonitrile and fractionated over a ZORBAX extended C18 column (Agilent, 5 μm particles, 4.6 mm ID and 250 mm in length). Peptides were separated on a 75 min linear gradient from 5% to 35% acetonitrile in 10 mM ammonium bicarbonate at a flow rate of 0.5 mL/min on an Agilent 1260 Infinity pump equipped with a degasser and a diode array detector (set at 214, 220, and 254 nm wavelength) from Agilent Technologies (Waldbronn, Germany). The samples were fractionated into a total of 96 fractions and then consolidated into 12 as described previously (*Edwards and Haas, 2016*). Samples were dried down under vacuum and reconstituted in 4% acetonitrile/5% formic acid for LC-MS/MS processing.

Peptides were analyzed on an Orbitrap Fusion Lumos mass spectrometer (Thermo Fisher Scientific) coupled to an Easy-nLC (Thermo Fisher Scientific). Peptides were separated on a microcapillary column (100 μm internal diameter, 25 cm long, filled using Maccel C18 AQ resin, 1.8 μm, 120A; Sepax Technologies). The total LC-MS run length for each sample was 180 min comprising a 165 min gradient from 6% to 30% acetonitrile in 0.125% formic acid. The flow rate was 300 nL/min and the column was heated to 60°C.

Data-dependent acquisition (DDA) mode was used for mass spectrometry data collection. A high resolution MS1 scan in the Orbitrap (m/z range 500–1,200, 60 k resolution, AGC $5 \times 105$, max injection time 100 ms, RF for S-lens 30) was collected from which the top 10 precursors were selected for MS2 analysis followed by MS3 analysis. For MS2 spectra, ions were isolated using a 0.5 m/z window using the mass filter. The MS2 scan was performed in the quadrupole ion trap (CID, AGC $1 \times 104$, normalized collision energy 30%, max injection time 35 ms) and the MS3 scan was analyzed in the Orbitrap (HCD, 60 k resolution, max AGC $5 \times 104$, max injection time 250 ms, normalized collision energy 50). For TMT reporter ion quantification, up to six fragment ions from each MS2 spectrum were selected for MS3 analysis using synchronous precursor selection (SPS).

Mass spectrometry data were processed using an in-house software pipeline (*Huttlin et al., 2010*). Raw files were converted to mzXML files and searched against a composite human uniprot

database (downloaded on 29th March 2017) containing sequences in forward and reverse orientations using the Sequest algorithm. Database searching matched MS/MS spectra with fully tryptic peptides from this composite dataset with a precursor ion tolerance of 20 ppm and a product ion tolerance of 0.6 Da. Carbamidomethylation of cysteine residues (+57.02146 Da) and TMT tags of peptide N-termini (+229.162932 Da) were set as static modifications. Oxidation of methionines (+15.99492 Da) was set as a variable modification. Linear discriminant analysis was used to filter peptide spectral matches to a 1% FDR (false discovery rate) as described previously (*Huttlin et al., 2010*). Non-unique peptides that matched to multiple proteins were assigned to proteins that contained the largest number of matched redundant peptides sequences using the principle of Occam's razor (*Huttlin et al., 2010*).

Quantification of TMT reporter ion intensities was performed by extracting the most intense ion within a 0.003 m/z window at the predicted m/z value for each reporter ion. TMT spectra were used for quantification when the sum of the signal-to-noise for all the reporter ions was greater than 200 and the isolation specificity was greater than 0.75 (*Ting et al., 2011*).

Base 2 logarithm of protein fold-changes were estimated by fitting a previously described Bayesian model (*O'Brien et al., 2018*) to the peptide level intensities. Protein estimates are reported as the mean of the posterior distribution for each parameter. Similarly, coefficients of variation are calculated by taking the posterior variance divided by the posterior mean. The probability of a small change ('P_null') was estimated as the frequency of posterior samples that fall within the interval $(-1,1)$ on the log2 scale.

## Polyubiquitination assay

Experiments were performed as per *Lu et al. (2014)*. One million 293Ts were plated in 6-well dishes overnight. Cells were transfected with 500 ng of plasmids containing CMV-6his-ubiquitin, CMV-3fl-EGFP, EF1A-DCAF15-3fl and/or CMV-SMC1A-HA. The SMC1A open reading frame was synthesized based on reference sequence NM_006306.3. After 36 hr, cells were treated with 2 μM indisulam and/or 10 μM MG132. 12 hr later, replicate wells were harvested for whole-cell lysates or for Ni-NTA pulldowns. Whole-cell lysates were made by extraction in 500 μl NP40 lysis buffer. For Ni-NTA pull-downs, cells were solubilized in 700 μl of guanidine buffer (6M guanidine-HCL, 0.1M $Na_2HPO_4$/ $NaH_2PO_4$, 10 mM imidazole, 0.05% TWEEN 20, pH 8.0), run through QIAshredder columns (Qiagen) and briefly sonicated. Purifications of 6his-ubiquinated species were performed as described in *Lu et al. (2014)*, except for the use of magnetic Ni-NTA beads (Thermo Fisher Scientific 88831) and the addition of 0.05% TWEEN 20 to wash buffers.

## Data processing

Unless otherwise specified, data were graphed and statistically analyzed using Prism (GraphPad Software). Sample size was not predetermined, No outliers were excluded. Unless otherwise noted, all data points represent biological replicates rather than technical replicates. We define 'technical replicates' as running an assay multiple times on the exact same sample.

## Acknowledgements

We thank Jonathan Paw for flow cytometry and cell sorting; Margaret Roy, Andrea Ireland, Twaritha Vijay, Nicole Fong for next-generation sequencing; Adam Baker and Matt Sooknah for RNA-seq data visualization; Hugo Hilton, David Stokoe, and Robert Cohen for critical reading of the manuscript, and members of the Calico Oncology group for helpful discussion.

## Additional information

### Competing interests

Jeffrey Settleman: Senior editor, *eLife*. Matthew F Pech, Linda E Fong, Jacqueline E Villalta, Leanne JG Chan, Samir Kharbanda, Jonathon J O'Brien, Fiona E McAllister, Ari J Firestone, Calvin H Jan: employee of Calico Life Sciences, LLC.

## Funding

| Funder | Author |
| --- | --- |
| Calico Life Sciences LLC | Matthew Pech |
| | Linda E Fong |
| | Jacqueline E Villalta |
| | Leanne JG Chan |
| | Samir Kharbanda |
| | Jonathon J O'Brien |
| | Fiona E McAllister |
| | Ari J Firestone |
| | Calvin H Jan |
| | Jeffrey Settleman |

The funders had no role in study design, data collection and interpretation, or the decision to submit the work for publication.

## Author contributions

Matthew F Pech, Conceptualization, Formal analysis, Investigation, Visualization, Methodology, Writing—original draft, Project administration, Writing—review and editing; Linda E Fong, Investigation, Visualization, Methodology, Writing—review and editing; Jacqueline E Villalta, Leanne JG Chan, Samir Kharbanda, Investigation; Jonathon J O'Brien, Software, Formal analysis, Writing—review and editing; Fiona E McAllister, Conceptualization, Supervision, Investigation; Ari J Firestone, Formal analysis, Investigation, Writing—review and editing; Calvin H Jan, Conceptualization, Supervision, Investigation, Writing—review and editing; Jeffrey Settleman, Conceptualization, Supervision, Project administration, Writing—review and editing

## Author ORCIDs

Matthew F Pech (iD) https://orcid.org/0000-0002-1451-0234
Jonathon J O'Brien (iD) http://orcid.org/0000-0001-9660-4797
Calvin H Jan (iD) http://orcid.org/0000-0001-9033-4028
Jeffrey Settleman (iD) https://orcid.org/0000-0003-3569-6493

## Decision letter and Author response

Decision letter https://doi.org/10.7554/eLife.47362.035
Author response https://doi.org/10.7554/eLife.47362.036

# Additional files

## Supplementary files

• Supplementary file 1. Design of genome-scale CRISPR library. sgRNA sequences and coordinates of the intended target locus are provided.
DOI: https://doi.org/10.7554/eLife.47362.019

• Supplementary file 2. NK CRISPR screen data. Normalized sgRNA counts and MAGeCK analysis output are provided.
DOI: https://doi.org/10.7554/eLife.47362.020

• Supplementary file 3. Raw MHC-I screen data. Normalized protospacer counts and MAGeCK analysis output are included.
DOI: https://doi.org/10.7554/eLife.47362.021

• Supplementary file 4. List of differentially expressed genes determined by RNA-seq of control, DCAF15 or PTPN2 KO K562 cells.
DOI: https://doi.org/10.7554/eLife.47362.022

• Supplementary file 5. Comparison of biotinylated proteins recovered from K562 cells expressing DCAF15-BIoID or GFP-BioID using isobaric labeling and mass spectrometry.
DOI: https://doi.org/10.7554/eLife.47362.023

• Supplementary file 6. List of sgRNA sequences used.
DOI: https://doi.org/10.7554/eLife.47362.024

• Supplementary file 7. Primer design for sequencing sgRNA libraries.

DOI: https://doi.org/10.7554/eLife.47362.025

• Supplementary file 8. List of antibodies used.
DOI: https://doi.org/10.7554/eLife.47362.026

• Transparent reporting form
DOI: https://doi.org/10.7554/eLife.47362.027

## Data availability

Sequencing data have been deposited in GEO under accession code GSE134173. All data generated or analyzed during this study are included in the manuscript and supporting files. Figure 1C: Supplementary File 2. Figure 2D: Supplementary File 3. Figure 4F: Supplementary File 4. Figure 7C: Supplementary File 5.

The following dataset was generated:

| Author(s) | Year | Dataset title | Dataset URL | Database and Identifier |
|---|---|---|---|---|
| Pech M, Settleman J | 2019 | Systematic identification of cancer cell vulnerabilities to natural killer cell-mediated immune surveillance | https://www.ncbi.nlm.nih.gov/geo/query/acc.cgi?acc=GSE134173 | NCBI Gene Expression Omnibus, GSE134173 |

The following previously published datasets were used:

| Author(s) | Year | Dataset title | Dataset URL | Database and Identifier |
|---|---|---|---|---|
| Bolouri H, Farrar JE, Triche T Jr, Ries RE, Lim EL, Alonzo TA, Ma Y, Moore R, Mungall AJ, Marra MA, Zhang J, Ma X, Liu Y, Auvil JMG, Davidsen TM, Gesuwan P, Hermida LC, Salhia B, Capone S Ramsingh G, Zwaan CM, Noort S, Piccolo SR, Kolb EA, Gamis AS, Smith MA, Gerhard DS, Meshinchi S | 2018 | TARGET AML RNAseq and clinical data | https://portal.gdc.cancer.gov/projects/TARGET-AML | National Cancer Institute GDC Data Portal, TARGET-AML |
| Cancer Genome Atlas Research Network, Ley TJ, Miller C, Ding L, Raphael BJ, Mungall AJ, Robertson A, Hoadley K, Triche TJ Jr, Laird PW, Baty JD, Fulton LL, Fulton R, Heath SE, Kalicki-Veizer J, Kandoth C, Klco JM, Koboldt DC, Kanchi KL, Kulkarni S, Lamprecht TL, Larson DE, Lin L, Lu C, McLellan MD, McMichael JF, Payton J, Schmidt H, Spencer DH, Tomasson MH, Wallis JW, Wartman LD, Watson MA, Welch J, Wendl MC, Ally A, Balasundaram M, Birol I, Butterfield Y, Chiu R, Chu | 2013 | TCGA LAML RNAseq and clinical data | https://portal.gdc.cancer.gov/projects/TCGA-LAML | National Cancer Institute GDC Data Portal, TCGA-LAML |

A, Chuah E, Chun HJ, Corbett R, Dhalla N, Guin R, He A, Hirst C, Hirst M, Holt RA, Jones S, Karsan A, Lee D, Li HI, Marra MA, Mayo M, Moore RA, Mungall K, Parker J, Pleasance E, Plettner P, Schein J, Stoll D, Swanson L, Tam A, Thiessen N, Varhol R, Wye N, Zhao Y, Gabriel S, Getz G, Sougnez C, Zou L, Leiserson MD, Vandin F, Wu HT, Applebaum F, Baylin SB, Akbani R, Broom BM, Chen K, Motter TC, Nguyen K, Weinstein JN, Zhang N, Ferguson ML, Adams C, Black A, Bowen J, Gastier-Foster J, Grossman T, Lichtenberg T, Wise L, Davidsen T, Demchok JA, Shaw KR, Sheth M, Sofia HJ, Yang L, Downing JR, Eley G

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
