## [Decision Letter]

Thank you for submitting your article "Systematic identification of cancer cell vulnerabilities to natural killer cell-mediated immune surveillance" for consideration by *eLife*. Your article has been reviewed by three peer reviewers, and the evaluation has been overseen by a Reviewing Editor and Tadatsugu Taniguchi as the Senior Editor. The reviewers have opted to remain anonymous.

The reviewers have discussed the reviews with one another and the Reviewing Editor has drafted this decision to help you prepare a revised submission.

Summary:

The authors employed a genome-wide CRISPR-Cas9 pooled sgRNA screen to identify genes that impact the susceptibility of the K562 cell line to NK-cells in a co-culture assay. Following infection with a genome-scale single guide RNA (sgRNA) library, Cas9-expressing K562 cells were challenged with NK-cell line in vitro or left to grow without challenge. A population of leukemic cancer cells that survived the NK-cell challenge, as well as control cells, were assessed for sgRNA abundance. This analysis revealed two sets of genes, inactivation of which either significantly enhanced or diminished the sensitivity of K562 cell line to NK-cell challenge. These two gene sets were comprised of genes involved in (1) formation of tumor immune synapses or (2) IFNγ signaling pathway. The most significantly enriched target in K562 population following NK-cell challenge was ICAM1, whereas the most significantly depleted target in K562 population following NK-cell challenge was the cullin4-RING E3 ubiquitin ligase (CRL4) substrate adaptor DCAF15. Further experiments confirmed that the disruption of this protein led to an increase in activating molecules such as CD80, but also inhibitory molecules such as MHC I. Since the effector cells were NK-92, the investigators saw an overall increase in NK-92 activity against the target cells, dependent on DCAF15 activity. Furthermore, the investigators presented data on novel substrates of DCAF15. They also provide first evidence between a link of DCAF15 and the cohesion complex. Ultimately, the study presented how the use of an unbiased CRISPR-Cas9 screen can lead to potential new therapies based on the discovery of unique proteins involved in immune surveillance.

Essential revisions:

1) All experiments including the CRISPR screen have been performed with NK92 cells, which do not recapitulate all features of primary NK cells. The authors should at least include primary NK cells to investigate the functional relevance of the proposed mechanism as validation (experiments shown in Figure 3B-C, Figure 4C, 4E). NK-92 cells are very different from primary fresh or activated NK cells, and therefore modulating the expression of DCAF15 may not even be relevant in clinical settings. This would increase the validity of the suggestion that DCAF inhibition may have immunomodulatory properties. The authors claim that the DCAF15 KO phenotype is dependent on CD80 upregulation and a more APC like state of K562s, which promotes NK92 activation. This mechanism should be validated for primary NK cells. Is CD28 really expressed on human NK cells and is this mechanism then relevant? Can the authors speculate if DCAF15 may play a role in bona fide APCs for regulating NK cell activation during immune responses?

2) Authors validate that indisulam-induced CD80 upregulation can also be achieved in other cell lines in addition to K562s. It should be tested if CD80 upregulation by indisulam treatment and/or by DCAF15 KO also makes these additional cell lines (Daudi and Ku812 cells) more sensitive to NK92 killing. What about CD33? This is also a top hit and antibodies are available – can that be used as control?

3) The authors identify ICAM1 and components of the IFNγ pathway as top hits as genes whose loss promote resistance of K562s towards NK92-mediated attack. These findings contrast recent data available on bioRxiv from a similar K562 CRISPR screen performed with primary NK cells (Klein et al., 2019), where B7H6 (NCR3LG1) was found as a single dominant hit in a comparable setting. B7H6 however only scored as #26 in their screen. This may indicate differences in K562-related resistance mechanisms against the functionally restricted NK cell line NK92 in contrast to primary NK cells. This must be at least discussed and further emphasizes the necessity to validate the outcome of the study with NK92 in a primary NK cell setting.

4) PTPN2 and DCAF15 knockout results in upregulation of MHC-I (Figure 4A), but still enhances NK cell killing, which is counterintuitive and against the dogma that low MHCI promotes NK cell cytotoxicity (missing self). This discrepancy should at least be discussed in more detail. Please also include a discussion on Dufva et al., 2018, where the expected effect that loss of IFN signaling enhanced tumor cell lysis is described. NK92 cells have restricted KIR expression, this may explain the reduced sensitivity towards MHC-I-mediated inhibition. Under this NK92 specific conditions further NK-cell activating effects of IFNγ may dominate in contrast to the situation in primary NK cells. These potential differences between NK92 and primary NK cells must be validated in more depth.

5) The authors do not provide/discuss a potential mechanism of how loss of IFNγ responsiveness in their screen promotes NK cell resistance. They exclude a direct cytostatic/cytotoxic effect of IFNγ. May that be related to a suppression of the proposed APC-like state of K562, associated with lower CD80 etc. levels? CD80 and other APC activation markers should be investigated e.g. on STAT1-KO (or IFNGR KO) K562 cells.

6) Can the RNA-seq data be exploited to speculate about the underlying mechanism of the growth inhibitory effect of PTPN2 knockout cells in contrast to WT K562 and DCAF15 KO cells in presence of IFNγ (Figure 4C)? Is there a threshold of hyperactivity of IFNγ signaling or may other PTPN2-controlled mechanisms play a role? Do the growth inhibitory effects of PTPN2-KO in presence of IFNγ produced by NK92 account for depletion of PTPN2-KO cells in the screen? Or does PTPN2 KO similar to DCAF15 KO also contribute to an APC-like inflamed state of K562s (e.g. CD80 upregulation)?

7) Figure 4B: The pSTAT1 levels are over-exposed in a manner that would prevent detection of any differences. I do see a reduced pSTAT1 phosphorylation of dCAF15 lanes versus control. Please repeat and show lower exposures – there may be something hidden.

8) Figure 7 is of great interest regarding the cohesion complex association. Can the authors explore that in more depth? What is the relation between cohesion mutations and DCAF165 expression? Can that at least be explored in silico in the AML samples? As it is the biochemical data appear a bit "lost".

9) Since the authors have identified many other intermediate molecules involved, it remains unclear what is the role of other receptors which are differentially regulated after inactivation of DCAF15. Some of these molecules may also be important in co-stimulation. Is there any specific reason why the authors focus on CD80 except that indisulam, an inhibitor of DCAF15, has a substantial impact on this receptor? Also, the ultimate mechanism by which DCAF15 disruption led to increased expression of CD80 is not clear.

10) As suggested by authors, upregulation of CD80 in DCAF15 KO cells may result in their differentiation towards APC-like properties. Indeed, they have shown that DCAF15 KO cells revealed higher levels of APC markers CD80, CD40, as well as MHC-I molecules which could give them the capacity to prime and present antigens to T-cells. Do T-lymphocytes have any role in better control of cancer cells lacking DCAF15? The authors have shown that the level of DCAF15 correlates with survival rate in patients, but this is by no means a proof of survival association with NK-cells.

---

## [Author Response]

Essential revisions:1a) All experiments including the CRISPR screen have been performed with NK92 cells, which do not recapitulate all features of primary NK cells. The authors should at least include primary NK cells to investigate the functional relevance of the proposed mechanism as validation (experiments shown in Figure 3B-C, Figure 4C, 4E). NK-92 cells are very different from primary fresh or activated NK cells, and therefore modulating the expression of DCAF15 may not even be relevant in clinical settings. This would increase the validity of the suggestion that DCAF inhibition may have immunomodulatory properties.

We agree that performing experiments with human PBMC-derived NK cells is important. While NK-92 cells display hallmark functions of conventional NK cells, including the rapid production of IFNγ following stimulation and the ability to mediate cellular cytotoxicity Freud et al., 2017, it is reasonable to expect a subset of the biology discovered in NK-92 screens will be NK-92 specific. We do note that many of the strongly scoring sensitization (NECTIN2) and resistance (ICAM1, B7H6, CD58, CD84) factors we discovered using NK-92 cells have previously been shown to modulate primary NK function (see Marcus et al., 2014; Brandt et al., 2009; Selvaraj et al., 1987; Rolle et al., 2016; Martin et al., 2001; Veillette, 2006; Wang et al., 2010; Stanietsky et al., 2009; Bottino et al.,), suggesting the screening system is useful for uncovering NK biology.

To address the reviewer’s comment, we tested NK cells derived from 6 different human donors. NK cells were isolated from PBMCs by negative selection, activated by 1000U/ml IL-2 overnight and challenged in competitive co-cultures with various K562 KO cell genotypes. Poststimulation, peripheral NK cells were >97% CD56+ CD3- and displayed markers of activation (CD69+ CD25+).

In this context, disruption of DCAF15 or PTPN2 promoted sensitization to primary NK cells, albeit with weaker effect compared to NK-92 cells. ICAM1 disruption promoted resistance to NK cell attack but did not confer complete protection. The effect of STAT1 disruption varied across donors, with STAT1 KO K562 cells strongly preferentially killed by primary NK cells from a subset of donors.

We also assessed the extent to which K562 DCAF15 KO cells promoted primary NK cell triggering. We found that 3 out of 6 donors exhibited significantly more degranulation when challenged by DCAF15 KO cells rather than control cells. Altogether, these findings support the relevance of DCAF15 inhibition as an immunomodulatory target.

We have revised the manuscript to incorporate the data from primary NK cells (Figure 3E-F and Figure 3—figure supplement 2). We have also explicitly stated in the manuscript which experiments were performed with NK-92 cells and which were performed with primary NK cells.

1b) The authors claim that the DCAF15 KO phenotype is dependent on CD80 upregulation and a more APC like state of K562s, which promotes NK92 activation. This mechanism should be validated for primary NK cells. Is CD28 really expressed on human NK cells and is this mechanism then relevant?

There is extensive literature support for the enhanced primary NK cell killing of cells transduced with CD80 or CD86 Townsend and Allison 1993; Chen et al., 1992; Wilson et al. 1999; Galea-Lauri et al., 1999; Chambers, Salcedo and Ljunggren, 1996; Azuma etal., 1992; Martin-Fontecha et al., 1999). In contrast, the role of CD28 in this process is controversial, with suggestions that CD28-dependent and -independent mechanisms may be involved. To address this question, we assayed the effect of CD28 agonism on NK-92 triggering in the presence of K562 target cells. The addition of CD28 agonistic antibodies to the co-cultures did not alter NK-92 degranulation (see Author response image 1), suggesting that our observations are CD28 independent. In figure 5—figure supplement 1E of our original manuscript, we provided flow cytometry data of CD28 expression on NK-92 cells. We have since examined CD28 expression in naïve and IL-2 stimulated human PBMC-derived NK cells using two different antibodies (clones 28.2, L293). We could not detect CD28 expression using either antibody clone within the CD56+ CD3- population (see Author response image 2 and data not shown). Both antibodies detected CD28 expression within the CD3+ population. Consistent with these findings, analysis of RNA-seq datasets from PBMC-derived FACS-sorted immune cell subtypes (Monaco et al., 2019) suggest CD28 mRNA is weak to absent in NK cells (see Author response image 3).

These results are consistent with the notion that NK-92 and primary NK cells are transducing CD80 stimuli through a CD28-independent mechanism. We have added CD28 immunophenotyping of NK cells to Figure 5—figure supplement 1F in the revised manuscript.

**Author response image 1. respfig1:** Effect of CD28 agonism on NK-92 degranulation.

**Author response image 2. respfig2:** CD28 expression in untouched PBMCs.

**Author response image 3. respfig3:** RNA-seq analysis of CD28 expression in isolated peripheral blood subtypes.

1c) Can the authors speculate if DCAF15 may play a role in bona fide APCs for regulating NK cell activation during immune responses?

Cross-talk between dendritic cells and NK cells is a well-established phenomenon that mutually regulates both cell types (Long, 2007; Vivier et al., 2008; Ferlazzo and Munz, 2004). These interactions often occur at sites of inflammation or secondary lymphoid organs and can promote maturation or lysis of the DCs, as well as priming or activation of the NK cells. It has been reported that NK cells facilitate lysis of immature/ nonactivated DCs from inflamed tissue, while activated DCs remain protected due to increased HLA-E expression.

Given that this manuscript is the first to explore the cellular functions of DCAF15, it is difficult to articulate a precise role of the protein in regulating NK-APC interactions. Generating conditional DCAF15-knockout mouse models and DCAF15 antibodies would be greatly beneficial for further exploring these areas of biology. To see whether DCAF15 was expressed in human dendritic cells, we examined RNA-seq datasets from PBMC-derived FACS-sorted immune cell subtypes (Monaco et al., 2019). DCAF15 was broadly expressed across PBMC subtypes, including pDCs and mDCs (see Author response image 4). Furthermore, in an independently collected public dataset (Oon et al., 2016), we note that treatment of DCs with TLR agonists was associated with an upregulation of DCAF15 expression (see Author response image 5), suggesting that DCAF15 activity may be modulated during DC activation.

We have added to the discussion the potential role of DCAF15 in the activation of APCs and their interactions with NK cells.

**Author response image 4. respfig4:** RNA-seq analysis of DCAF15 expression in isolated peripheral blood subtypes.

**Author response image 5. respfig5:** DCAF15 expression in human pDCs after stimulation with TLR agonists.

2a) Authors validate that indisulam-induced CD80 upregulation can also be achieved in other cell lines in addition to K562s. It should be tested if CD80 upregulation by indisulam treatment and/or by DCAF15 KO also makes these additional cell lines (Daudi and Ku812 cells) more sensitive to NK92 killing.

We agree that extending our observations on DCAF15 to other cell lines is important. Given the uncharacterized nature of DCAF15, we focused on understanding its role in K562 cells before embarking on experiments in other cellular contexts.

To address this point, we engineered Daudi cell lines with stable Cas9 expression, and then infected with them sgRNAs against DCAF15, PTPN2, STAT1 or ICAM1. Western blot, flow cytometry and RNA-seq confirmed a high level of target depletion (data added as Figure 3—figure supplement 1D-F). Consistent with our findings in K562 cells, competitive co-culture assays revealed that disruption of ICAM1 in Daudi cells was highly protective against NK-92 cell killing, whereas disruption of DCAF15 or PTPN2 enhanced killing (data added as Figure 3D).

We attempted to also extend findings to Ku812 cells. However, these cells grew poorly after Cas9 infection, so we could not proceed with further experimentation.

2b) What about CD33? This is also a top hit and antibodies are available – can that be used as control?

As the reviewer notes, RNA-seq analysis suggests that CD33 is transcriptionally upregulated in K562 DCAF15 KO cells (Figure 4F). CD33 is a classic myeloid antigen, with high expression in myeloid progenitors as well as tissue macrophages and myeloid dendritic cells(Crocker et al., 2001). We note that neither the published literature nor our NK-92 CRISPR screening data support the notion that expression of CD33 is an important factor modulating NK activity. We measured CD33 by flow cytometry and detected a small but reproducible increase in CD33 levels between control and DCAF15 KO K562 cells (see Author Response Image 6; 1.15-fold more CD33, P=0.024 Mann-Whitney test).

**Author response image 6. respfig6:** CD33 expression in K562 cells of indicated genotypes.

3) The authors identify ICAM1 and components of the IFNγ pathway as top hits as genes whose loss promote resistance of K562s towards NK92-mediated attack. These findings contrast recent data available on bioRxiv from a similar K562 CRISPR screen performed with primary NK cells (Klein et al., 2019), where B7H6 (NCR3LG1) was found as a single dominant hit in a comparable setting. B7H6 however only scored as #26 in their screen. This may indicate differences in K562-related resistance mechanisms against the functionally restricted NK cell line NK92 in contrast to primary NK cells. This must be at least discussed and further emphasizes the necessity to validate the outcome of the study with NK92 in a primary NK cell setting.

Revision 1a addresses the relevance of DCAF15 KO, PTPN2 KO an ICAM1 KO in a primary NK cell setting.

Klein et al., performed a CRISPR co-culture screen with PBMC-derived NK cells and K562 cells. In their screening format, CRISPR-mutagenized K562 cells were challenged twice with NK cells, exerting a very high selective pressure against the K562 cells. Such an experimental design is only able to discover resistance factors, not sensitization factors, and likely drives differences in screening outcomes much more than the use of primary NK cells vs. NK-92 cells.

Our screens, as well as those in Klein et al., revealed that disruption of B7H6 is a strongly scoring resistance mechanism. B7H6 was the only hit recovered in the Klein et al., screen, with roughly two-thirds of the surviving cells from the CRISPR screen being B7H6-negative, and B7H6 guides enriched roughly 8-fold at the end of their screen. As the authors mention in their discussion, the absence of additional hits in the screen reflected the design of their assay, rather than the absence of other NK-resistance mechanisms in K562 cells.

We note that Klein et al., analyzed their screening data manually, without using any statistical techniques designed for the analysis of CRISPR screens. The dataset has not been made available with the manuscript, so we cannot determine for ourselves what resistance mechanisms may have gone unappreciated. We also note that Klein et al., compared sgRNA abundance in cells immediately post-sgRNA infection to cells after the second round of NK challenge. As such, changes in sgRNA abundance reflect both gene essentiality during normal cell culture as well as modulation of NK sensitivity. In our screens, we controlled for any gene dropout over time by comparing sgRNA abundance in K562 cells +/- NK92 challenge.

4a) PTPN2 and DCAF15 knockout results in upregulation of MHC-I (Figure 4A), but still enhances NK cell killing, which is counterintuitive and against the dogma that low MHCI promotes NK cell cytotoxicity (missing self). This discrepancy should at least be discussed in more detail. NK92 cells have restricted KIR expression, this may explain the reduced sensitivity towards MHC-I-mediated inhibition. Under this NK92 specific conditions further NK-cell activating effects of IFNγ may dominate in contrast to the situation in primary NK cells. These potential differences between NK92 and primary NK cells must be validated in more depth.

In revision 1a, we observed that DCAF15 or PTPN2 promoted sensitization to primary NK cells. The effect was weaker compared to NK-92 cells, especially for DCAF15 KO cells. As the reviewer notes, NK-92 have an immunophenotype reminiscent of unlicensed NK cells (CD3- CD56-bright; CD16low/neg), with a limited KIR expression compared to primary NK cells (Maki et al., 2001). The reduced ability of NK-92 cells to transduce MHC-I inhibitory signals likely explains why we observe a more potent sensitizing effect of DCAF15 or PTPN2 disruption in that setting. We do note that the absolute level of MHC-I expression in DCAF15 KO and PTPN2 KO K562 cells remains relatively low, and that we have not characterized which MHC-I subclasses are driving the expression difference. We have added to the revised manuscript a discussion of the “MHC discrepancy.”

4b) Please also include a discussion on Dufva et al., 2018, where the expected effect that loss of IFN signaling enhanced tumor cell lysis is described.

The reviewer does not cite a scientific article, but rather an abstract presented at a conference, which includes no figures or primary data. According to the Abstract, Dufva et al., used CRISPR screening to find that disruption of the IFN-JAK-STAT signaling pathway sensitizes various cancer cell lines to primary NK cell effector functions (Dufva et al., 2018). They suggest that MHC-I upregulation in response to NK cell-derived IFNγ inhibits NK activation and reduces cell lysis.

Our data in Figure 3 is consistent with the notion that target cell MHC-I expression and effector cell KIR expression are important cell surface factors regulating target cell lysis. The outcome of cancer cell IFNγ signaling is variable, and can include MHC-I upregulation, growth suppression, and/or immuno-modulation. We observe that whether cancer cell IFNγ signaling ultimately leads to reduced or enhanced NK cell lysis depends on the exact target and effector populations studied.

5) The authors do not provide/discuss a potential mechanism of how loss of IFNγ responsiveness in their screen promotes NK cell resistance. They exclude a direct cytostatic/cytotoxic effect of IFNγ. May that be related to a suppression of the proposed APC-like state of K562, associated with lower CD80 etc. levels? CD80 and other APC activation markers should be investigated e.g. on STAT1-KO (or IFNGR KO) K562 cells.

Initially, we studied the effect of STAT1 disruption in K562 cells when challenged with NK-92 cells. We have since examined the effect of STAT1 KO in more cellular contexts. We found that STAT1 disruption had a highly variable effect, promoting resistance or sensitization to NK cells in a target cell and effector cell-dependent manner (data added to Figure 3D-E). Rebuttal 4B describes how this variability likely arises.

We immunophenotyped K562 STAT1 KO cells before and after 24hours of IFNγ treatment. In the basal state, MHC-I, MHC-II, CD80 and CD40 and ICAM1 levels were either unchanged in STAT1 KO cells or modestly decreased (Author response image 7). Of these markers, MHC-I, CD40 and ICAM1 were increased after IFNγ stimulation in a STAT1-dependent manner. These data suggest that changes in APC activation markers are not responsible for promoting NK-92 resistance. We hypothesize that STAT1 KO K562 cells exhibited NK-92 resistance due to impaired IFNγ-induced ICAM1 upregulation and the relatively limited ability of NK-92 cells to receive MHC-I inhibitory signals.

We have added a subset of the STAT1 KO immunophenotype data to Figure 4—figure supplement 1C in the revised manuscript.

**Author response image 7. respfig7:** Expression of APC activation markers in STAT1 KO K562 cells.

6) Can the RNA-seq data be exploited to speculate about the underlying mechanism of the growth inhibitory effect of PTPN2 knockout cells in contrast to WT K562 and DCAF15 KO cells in presence of IFNγ (Figure 4C)? Is there a threshold of hyperactivity of IFNγ signaling or may other PTPN2-controlled mechanisms play a role? Do the growth inhibitory effects of PTPN2-KO in presence of IFNγ produced by NK92 account for depletion of PTPN2-KO cells in the screen? Or does PTPN2 KO similar to DCAF15 KO also contribute to an APC-like inflamed state of K562s (e.g. CD80 upregulation)?

IFNγ treatment is growth-suppressive in PTPN2 KO cells, but is not directly cytostatic/cytotoxic to control K562 or DCAF15 KO cells (Figure 4C). Normal K562 cells respond transcriptionally to IFNγ by dramatically upregulating antiviral genes and components of the antigen processing and presentation pathway (Figure 4—figure supplement 1B). In contrast, PTPN2 KO cells were enriched for inflammation and interferon-associated Gene Ontology (GO) terms in the unstimulated state (Figure 4—figure supplement 1D). After IFNγ treatment, PTPN2 KO cells had exaggerated induction of IFNγ-responsive genes and were also enriched for apoptotic GO gene categories (Figure 4—figure supplement 1E). Thus, it appears that loss of appropriate negative feedback on IFNγ signaling may promote cell death, contributing to the observed growth inhibitory effect.

It is unclear whether the IFNγ-induced growth suppression of PTPN2 KO cells fully accounts for the sensitivity of these cells to NK cell lysis. PTPN2 KO cells are immuno-modulated in a way expected to both increase (higher ICAM1) and decrease (higher MHC-I) NK cell activation.

We have added a volcano plot showing the transcriptional response to IFNγ to Figure 4—figure supplement 1B. We have added PTPN2 immunophenotypic changes to Figure 4—figure supplement 1C. In the Discussion section, we devote more space to discussing the PTPN2 findings.

7) Figure 4B: The pSTAT1 levels are over-exposed in a manner that would prevent detection of any differences. I do see a reduced pSTAT1 phosphorylation of dCAF15 lanes versus control. Please repeat and show lower exposures – there may be something hidden.

We have included a pSTAT1 western blot with a reduced exposure in Figure 4B in the revised manuscript.

We note that these experiments were performed three independent times, and we never detected consistently altered levels of pSTAT1, STAT1, JAK1/2 or IFNGR1 in DCAF15 KO cells before or after IFNγ stimulation. We also note that the RNAseq data of DCAF15 KO cells after IFNγ treatment (Figure 4) is consistent with the notion that the transcriptional response of DCAF15 KO cells to IFNγ is not dramatically dysregulated.

8) Figure 7 is of great interest regarding the cohesion complex association. Can the authors explore that in more depth? What is the relation between cohesion mutations and DCAF165 expression? Can that at least be explored in silico in the AML samples? As it is the biochemical data appear a bit "lost".

We have examined the relationship between cohesin-family mutations and DCAF15 expression levels in AML. DCAF15 mRNA expression trended lower in cohesin wild-type AML samples, though the differences were not significant (see Author response image 8).

In the original submission, we had detected interactions between exogenous DCAF15-BioID constructs and endogenous SMC proteins. We have since strengthened these observations by performing in vivo ubiquitination assays. We found that DCAF15 promotes the ubiquitination of SMC1 and that the level of SMC ubiquitination by DCAF15 was blunted by indisulam cotreatment. These data strengthens our conclusion that cohesin complex proteins are bona fide substrates of DCAF15. We have added these data to Figure 7F in the revised manuscript.

**Author response image 8. respfig8:** DCAF15 expression in TCGA AML samples stratified by cohesin mutation status.

9) Since the authors have identified many other intermediate molecules involved, it remains unclear what is the role of other receptors which are differentially regulated after inactivation of DCAF15. Some of these molecules may also be important in co-stimulation. Is there any specific reason why the authors focus on CD80 except that indisulam, an inhibitor of DCAF15, has a substantial impact on this receptor? Also, the ultimate mechanism by which DCAF15 disruption led to increased expression of CD80 is not clear.

Our initial interest on CD80 arose because it was one of the most differentially expressed genes in K562 DCAF15 KO cells compared to control KO cells (Figure 4F). After validating this observation, we then discovered that indisulam could modulate CD80 expression to a similar extent as genetic depletion of DCAF15.

As the reviewer mentions, it is possible for other important co-stimulatory receptors to be differentially expressed in DCAF15 KO cells. We addressed this possibility in Figure 5—figure supplement 1A by measuring the expression level of a variety of other receptors implicated by the NK screen and/or in the literature to see whether they were altered in DCAF15 KO cells. In the original manuscript, we included data on ULBP2/5/6, CD58, NECTIN2, ICAM1 and IFNGR1. We have now also included in Figure 5—figure supplement 1A data on B7H6/NCR3LG1 and have shown that its levels are also unaltered in DCAF15 KO cells.

The discussion and figure 7G lay out our speculative model for how DCAF15 disruption ultimately leads to changes in CD80 expression. We hypothesize that cohesin proteins are direct client substrates of DCAF15, and that in the absence of DCAF15, cohesin dysregulation leads to differentiation of K562 cells towards an APC-like fate, which includes the upregulation of CD80.

10) As suggested by authors, upregulation of CD80 in DCAF15 KO cells may result in their differentiation towards APC-like properties. Indeed, they have shown that DCAF15 KO cells revealed higher levels of APC markers CD80, CD40, as well as MHC-I molecules which could give them the capacity to prime and present antigens to T-cells. Do T-lymphocytes have any role in better control of cancer cells lacking DCAF15? The authors have shown that the level of DCAF15 correlates with survival rate in patients, but this is by no means a proof of survival association with NK-cells.

As the reviewer notes, the immunophenotypic changes of DCAF15 KO cells could conceivably enhance the ability of tumor cells to directly prime T cell responses. Further experimentation would be needed to support such a claim. We agree that the correlation between DCAF15 expression and AML survival is not proof of enhanced NK-mediated control of AML blasts in DCAF15-low patients. In the manuscript, we simply state that “lower DCAF15 function, achieved pharmacologically or by genetic means, is associated with favorable immunophenotypes in vitro and improved outcomes in AML patients”. An intriguing observation is that AML patients receiving allogenic stem cell transplants from KIR-mismatched donors have enhanced graft vs. leukemia activity and reduced relapse (Ruggeri et al., 2002), suggesting that NK cells can have clinically meaningful activity against AML.